# When to Trust Aggregated Gradients: Addressing Negative Client Sampling in Federated Learning

## Abstract

Federated Learning has become a widely-used framework which allows learning a global model on decentralized local datasets under the condition of protecting local data privacy. However, federated learning faces severe optimization difficulty when training samples are not independently and identically distributed (non-i.i.d.). In this paper, we point out that the client sampling practice plays a decisive role in the aforementioned optimization difficulty. We find that the negative client sampling will cause the merged data distribution of currently sampled clients heavily inconsistent with that of all available clients, and further make the aggregated gradient unreliable. To address this issue, we propose a novel learning rate adaptation mechanism to adaptively adjust the server learning rate for the aggregated gradient in each round, according to the consistency between the merged data distribution of currently sampled clients and that of all available clients. Specifically, we make theoretical deductions to find a meaningful and robust indicator that is positively related to the optimal server learning rate and can effectively reflect the merged data distribution of sampled clients, and we utilize it for the server learning rate adaptation. Extensive experiments on multiple image and text classification tasks validate the great effectiveness of our method.

## 1 Introduction

As tremendous data is produced in various edge devices (e.g., mobile phones) every day, it becomes important to study how to effectively utilize the data without revealing personal information and privacy. Federated Learning (Konečný et al., 2016; McMahan et al., 2017) is then proposed to allow many clients to jointly train a well-behaved global model without exposing their private data. In each communication round, clients get the global model from a server and train the model locally on their own data for multiple steps. Then they upload the accumulated gradients only to the server, which is responsible to aggregate (average) the collected gradients and update the global model. By doing so, the training data never leaves the local devices.

It has been shown that the federated learning algorithms perform poorly when training samples are not independently and identically distributed (non-i.i.d.) across clients (McMahan et al., 2017; Li et al., 2021), which is the common case in reality. Previous studies (Zhao et al., 2018; Karimireddy et al., 2020) mainly attribute this problem to the fact that the non-i.i.d. data distribution leads to the divergence of the directions of the local gradients. Thus, they aim to solve this issue by making the local gradients have more consistent directions (Li et al., 2018; Sattler et al., 2021; Acar et al., 2020). However, we point out that the above studies overlook the negative impact brought by the client sampling procedure (McMahan et al., 2017; Fraboni et al., 2021b), whose existence we think should be the main cause of the optimization difficulty of the federated learning on non-i.i.d. data.

Client sampling is widely applied in the server to solve the communication difficulty between the great number of total clients and the server with limited communication capability, by only sampling a small part of clients to participate in each round. We find that the client sampling induces the negative effect of the non-i.i.d. data distribution on federated learning. For example, assume each client performs one full-batch gradient descent and uploads the local full-batch gradient immediately (i.e., FedSGD in McMahan et al. (2017)), (1) if all clients participate in the current round, it is equivalent

to performing a global full-batch gradient descent on all training samples, and the aggregated server gradient is accurate regardless of whether the data is i.i.d. or not; (2) if only a part of clients is sampled for participation, the aggregated server gradient will deviate from the above global full-batch gradient and its direction depends on the merged data distribution of the currently sampled clients (i.e., the label distribution of the dataset constructed by data points from all sampled clients). The analysis is similar in the scenario where clients perform multiple local updates before uploading the gradients, and we have a detailed discussion in Appendix B.

The above analysis indicates that the reliability of the averaged gradients depends on the consistency between the merged data distribution of currently sampled clients and that of all available clients. Specifically, take the image classification task as an example[1] (refer to Figure 1): (1) if local samples from all selected clients are almost all cat images, the averaged gradient's direction deviates far away from the ideal server gradient averaged by all clients' local gradients. In this case, we may not trust the aggregated gradient, and decrease the server learning rate. (2) However, if the merged data distribution of selected clients matches well with the merged data distribution of all clients, the averaged gradient's direction is more reliable. Thus, we should relatively enlarge the server learning rate. This example motivates us to set dynamic server learning rates across rounds instead of a fixed one as previous methods do.

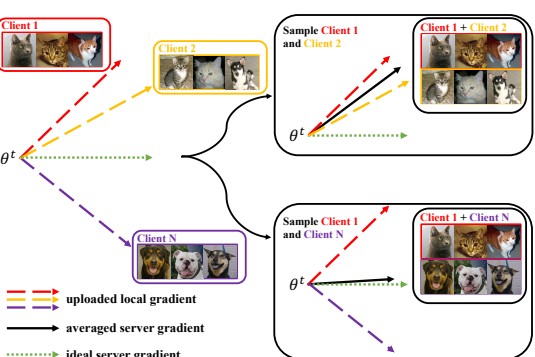

Figure 1: An illustration of the impact of client sampling through a binary (cats v.s. dogs) image classification task. **(Right Top)**: If Client 1 and Client 2 are selected, their merged local samples are almost all cat images. Then the averaged gradient deviates far away from the ideal gradient that would be averaged by all clients' local gradients. **(Right Bottom)**: If Client 1 and Client N are selected, their merged data distribution matches well with the global data distribution merged by all clients' data, and the averaged gradient has a more reliable direction.

In this paper, we first analyze the impact of client sampling and are motivated to mitigate its negative impact by dynamically adjusting the server learning rates based on the reliability of the aggregated server gradients. We theoretically show that the optimal server learning rate in each round is positively related to an indicator called the **Gradient Similarity–aware Indicator (GSI)**, which can reflect the merged data distribution of sampled clients by measuring the dissimilarity between uploaded gradients. Based on this indicator, we propose a gradient similarity–aware learning rate adaptation mechanism to adaptively adjust the server learning rates. Furthermore, our method will adjust the learning rate for each parameter group (i.e., weight matrix) individually based on its own $GSI$, in order to solve the issue of the inconsistent fluctuation patterns of $GSI$ across parameter groups and achieve more precise adjustments. Extensive experiments on multiple benchmarks show that our method consistently brings improvement to various state-of-the-art federated optimization methods in various settings.

## 2 RELATED WORK

Federated Averaging (FedAvg) is first proposed by McMahan et al. (2017) for the federated learning, and its convergence property is then widely studied (Li et al., 2019; Karimireddy et al., 2020). While FedAvg behaves well on the i.i.d. data with client sampling, further studies (Karimireddy et al., 2020) reveal that its performance on the non-i.i.d. data degrades greatly. Therefore, existing studies focus on improving the model's performance when the training samples are non-i.i.d. under the partial client participation setting. They can be divided into the following categories:

**Advancing the optimizer used in the server's updating:** Compared with FedAvg that is equivalent to applying SGD during the server's updating, other studies take steps further to use advanced server optimizers, such as using SGDM (Hsu et al., 2019; Wang et al., 2019b) and adaptive opti-

---

[1]Though in our example in Figure 1, for simplicity, we assume the merged data distribution of all available clients is balanced, we do not have this assumption in our later analysis.

mizers (Reddi et al., 2020) like Adam (Kingma & Ba, 2014). However, their improvement comes from the effects of using advanced server optimizers, and they do not truly target on addressing the problems of the non-i.i.d. data distribution and the client sampling. We show that our method can be effectively applied with these federated optimizers.

**Improving the directions of the averaged server gradients:** (1) Some studies (Wang et al., 2020; Yeganeh et al., 2020; Wu & Wang, 2021) manage to introduce better gradients' re-weighting procedures during server's aggregation than the naive averaging (McMahan et al., 2017). However, they either require an extra global dataset for validation or fail to tackle the situation where most of the sampled clients have similar but skewed data distributions. (2) Instead of focusing on the server side, other studies choose to improve the local training on the client side. For example, FedProx (Li et al., 2018) and FedDyn (Acar et al., 2020) choose to add the regularization terms to make local gradients' directions more consistent; SCAFFOLD (Karimireddy et al., 2020) and VRLSGD (Liang et al., 2019) send the historical aggregated gradients to each client for helping correct its local gradient's direction. In this paper, we consider from an orthogonal angle that aims to find an optimal server learning rate after the averaged gradient is determined. Thus, our method can be naturally combined with them.

**Introducing new client sampling strategies:** Being aware of the important role the client sampling plays in federated learning, some studies (Ribero & Vikalo, 2020; Chen et al., 2020; Cho et al., 2020; Fraboni et al., 2021a) try to propose better client sampling schemes than the default *uniform sampling without replacement strategy* (McMahan et al., 2017). They attempt to select the most important clients based on some statistics of their local gradients, such as the gradient's norm. Our work is also orthogonal to them and can bring additional improvement when applied with them.

## 3 METHODOLOGY

### 3.1 PROBLEM DEFINITION

The optimization target of the federated learning (McMahan et al., 2017) can be formalized as

$$\min_{\boldsymbol{\theta}} L(\boldsymbol{\theta}) = \min_{\boldsymbol{\theta}} \sum_{k=1}^{N} \frac{n_k}{n} L_k(\boldsymbol{\theta}), \tag{1}$$

where $N$ is the total number of clients, $n_k$ is the number of local training samples on client $k$, $n = \sum_{k=1}^{N} n_k$ is the total number of training samples, and $L_k(\boldsymbol{\theta}) = \frac{1}{n_k} \sum_{i=1}^{n_k} l_k(\boldsymbol{\theta}; \boldsymbol{x}_i^{(k)}, \boldsymbol{y}_i^{(k)})$ is the local training objective for client $k$.

The most popular federated learning framework is the FedAvg (McMahan et al., 2017), which broadcasts the current global model $\boldsymbol{\theta}^t$ to the available clients when the $(t+1)$-th round begins and allows each client to perform the multi-step centralized training on its own data. Then each client only sends the accumulated local gradient $\boldsymbol{g}_k^t$ ($k = 1, \cdots, N$) to the server, and the server will aggregate (average) the local gradients and update the global model as the following:

$$\boldsymbol{\theta}^{t+1} = \boldsymbol{\theta}^t - \eta_s \sum_{k=1}^{N} \frac{n_k}{n} \boldsymbol{g}_k^t, \tag{2}$$

where $\eta_s$ is the server learning rate. The server will send the $\boldsymbol{\theta}^{t+1}$ to clients again for the next round's training.

In most real cases, due to the great number of clients and the server's limited communication capability, the server will perform a *client sampling* strategy to sample a small portion of clients only in each round. Also, since we assume the number of local updates performed during each client's local training is the same in each training round following previous studies (McMahan et al., 2017; Reddi et al., 2020),[2] then all $n_k$ are assumed to be the same. Therefore, Eq. (2) can be re-written as

$$\boldsymbol{\theta}^{t+1} = \boldsymbol{\theta}^t - \eta_s \frac{1}{r} \sum_{k \in S(N,r;t)} \boldsymbol{g}_k^t, \tag{3}$$

where $r$ is the number of sampled clients, $S(N, r; t)$ represents a sampling procedure of sampling $r$ clients from $N$ clients in the $(t+1)$-th round, and we abbreviate it to $S^t$ in the following paper.

---

[2] Our later proposed method in Section 3.3 can also be applied in the setting where the number of local updates varies across clients, combining with specific optimization methods like FedNova (Wang et al., 2020).

## 3.2 ANALYSIS OF THE IMPACT OF CLIENT SAMPLING

As we can except, in a given round, the averaged gradient under partial client participation will deviate from the ideal gradient under full client participation, which we think has a reliable direction since it contains the data information of all clients. Thus, the motivation of most current studies can be summarized into optimizing the following problem:

$$\max_{\boldsymbol{g}^t}\{\texttt{CosSim}(\boldsymbol{g}^t,\boldsymbol{g}_c^t)\}, \tag{4}$$

where $\boldsymbol{g}^t = \frac{1}{r}\sum_{k\in S^t}\boldsymbol{g}_k^t$ and $\boldsymbol{g}_c^t = \frac{1}{N}\sum_{k=1}^N \boldsymbol{g}_k^t$. They achieve this goal indirectly from the perspective of improving the local gradients, either by adjusting the direction of $\boldsymbol{g}_k^t$ (Li et al., 2018; Karimireddy et al., 2020; Acar et al., 2020), or by sampling better groups of clients (Chen et al., 2020; 2021).

However, based on the analysis in Figure 1, we are motivated to mitigate the inconsistent between $\boldsymbol{g}^t$ and $\boldsymbol{g}_c^t$ from a different perspective, that is **scaling $\boldsymbol{g}^t$ to achieve the nearest distance between it and $\boldsymbol{g}_c^t$** (refer to Figure 2). By doing so, a great advantage is to **avoid updating the model towards a skewed direction overly when the uploaded gradients are similar but skewed** (refer to the right top of Figure 1). Then our optimization target can be written as:[3]

$$\eta_o^t = \arg\min_{\eta^t} \|\eta^t\boldsymbol{g}^t - \eta_s\boldsymbol{g}_c^t\|^2, \tag{5}$$

where $\eta_s$ is the optimal global learning rate used under the full client participation setting.

Moreover, we point out that since our method is orthogonal to previous studies, we can get a more comprehensive optimization target by combining Eq. (5) with Eq. (4) as:

$$\min_{\eta^t}\|\eta^t\boldsymbol{g}^t - \eta_s\boldsymbol{g}_c^t\|^2, \quad \text{s.t.} \quad \boldsymbol{g}^t = \arg\max_{\boldsymbol{g}}\{\texttt{CosSim}(\boldsymbol{g},\boldsymbol{g}_c^t)\}. \tag{6}$$

However, in the scope of this paper, we only focus on the first part of Eq. (6) that is overlooked by previous studies, while **assuming $\boldsymbol{g}^t$ is already obtained or optimized by existing methods**. Then, by solving Eq. (5) we can get

$$\eta_o^t = \eta_s\left(\langle\boldsymbol{g}^t,\boldsymbol{g}_c^t\rangle/\|\boldsymbol{g}^t\|^2\right), \tag{7}$$

where $<\cdot,\cdot>$ is an element-wise dot product operation.

In the $(t+1)$-th round given $\boldsymbol{\theta}^t$, $\boldsymbol{g}_c^t$ is the determined but unknown parameters, which means it is impossible to calculate $<\boldsymbol{g}^t,\boldsymbol{g}_c^t>$ precisely. However, we can still find the factors that will affect $\eta_o^t$ by analyzing its upper bound:

$$\eta_o^t \le \eta_s\frac{\|\boldsymbol{g}_c^t\|}{\|\boldsymbol{g}^t\|} = \frac{\eta_s\|\boldsymbol{g}_c^t\|}{\sqrt{\frac{1}{r}\sum_{k\in S_t}\|\boldsymbol{g}_k^t\|^2}}\frac{\sqrt{\frac{1}{r}\sum_{k\in S_t}\|\boldsymbol{g}_k^t\|^2}}{\|\boldsymbol{g}^t\|}. \tag{8}$$

In the first equality of Eq. (8), we make the further derivation to separate the **scale component** $\sqrt{\frac{1}{r}\sum_{k\in S_t}\|\boldsymbol{g}_k^t\|^2}$ that is decided by the scales of the local gradients, and the **direction component** $\|\boldsymbol{g}^t\|/\sqrt{\frac{1}{r}\sum_{k\in S_t}\|\boldsymbol{g}_k^t\|^2}$ that depends on the similarity between the directions of local gradients, from $\|\boldsymbol{g}^t\|$. Now in the first term of the final equation of Eq. (8), $\|\boldsymbol{g}_c^t\|$ can be considered as an unknown constant in this round given $\boldsymbol{\theta}^t$; the scale component $\sqrt{\frac{1}{r}\sum_{k\in S_t}\|\boldsymbol{g}_k^t\|^2}$ can also be regarded as a constant which is barely affected by the sampling results in current round, since we assume each client performs the same number of local updates, so the difference between the norms of all local gradients can be negligible (especially when the local optimizer is Adam),[4] compared with the difference of local gradients' directions. However, the second term $(\sqrt{\frac{1}{r}\sum_{k\in S_t}\|\boldsymbol{g}_k^t\|^2})/\|\boldsymbol{g}^t\|$ indeed depends on the merged data distribution of currently sampled clients, as the directions of the local gradients depends on the sampled clients' local data distributions. Since it measures the normalized similarity between local gradients, we name it the **Gradient Similarity–aware Indicator (GSI)**:

$$GSI^t = \sqrt{(\sum\nolimits_{k\in S_t}\|\boldsymbol{g}_k^t\|^2)/(r\|\boldsymbol{g}^t\|^2)}. \tag{9}$$

---

[3]In the following paper, $\|\cdot\|$ represents the Frobenius norm.

[4]We make the detailed discussion and provide the empirical evidence in Appendix C.

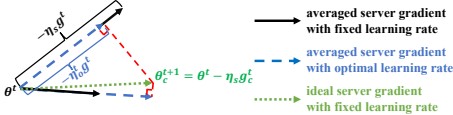

Figure 2: Illustration of our motivation.

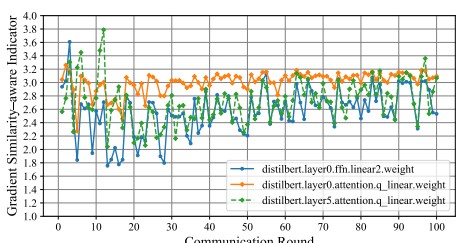

Figure 3: Fluctuation patterns of $GSI$s in different parameter groups when federated training on 20NewsGroups (Lang, 1995) with DistilBERT (Sanh et al., 2019).

---

**Algorithm 1** Gradient Similarity–Aware Learning Rate Adaptation for Federated Learning

---

**Server Input:** $\boldsymbol{\theta}^0$, $\eta_0$, $\beta = 0.9$, $\gamma = 0.02$
**for** each round $t = 0$ **to** $T$ **do**
    sample client $S^t \subset \{1, \cdots, N\}$
    **for** $k \in S^t$ **in parallel do**
        $\boldsymbol{g}_k^t \leftarrow$ **LocalTraining**$(k, \boldsymbol{\theta}^t, D_k)$
    **end for**
    $\boldsymbol{g}^t \leftarrow \frac{1}{r} \sum_{k \in S^t} \boldsymbol{g}_k^t$
    **for** each param. group $\boldsymbol{\theta}_P^t$'s gradient $\boldsymbol{g}_p^t$ **do**
        $GSI_P^t \leftarrow \sqrt{(\sum_{k \in S_t} \|\boldsymbol{g}_{P,k}^t\|^2)/(r\|\boldsymbol{g}_P^t\|^2)}$
        $B_P^0 \leftarrow GSI_P^0$
        $\tilde{\eta}_P^t \leftarrow \min\{\max\{GSI_P^t/B_P^t, 1 - \gamma t\}, 1 + \gamma t\}$
        $B_P^{t+1} \leftarrow \beta B_P^t + (1 - \beta)GSI_P^t$
        $\boldsymbol{g}_P^t \leftarrow \tilde{\eta}_P^t \boldsymbol{g}_P^t$
    **end for**
    $\boldsymbol{\theta}^{t+1} \leftarrow \boldsymbol{\theta}^t - \eta_0 \boldsymbol{g}^t$
**end for**

---

The reason why we do not choose $1/\|\boldsymbol{g}^t\|$ as the indicator and make the further derivation in Eq. (8) is, the scales of the gradients will naturally decreases when training goes on, and $1/\|\boldsymbol{g}^t\|$ will be in an increasing trend. This leads to the consistently increasing server learning rates, which makes SGD based optimization (which happens in the server aggregation phase) fail at the end of training. However, $GSI$ is a normalized indicator whose scale is stable across rounds.

We surprisingly find that the $GSI$ happens to have the similar form to the gradient diversity concept proposed in the distributed learning field (Yin et al., 2018), however, they have different functions: the gradient diversity concept is originally used to illustrate that allocating heterogeneous data to different nodes helps distributed learning; while in federated learning, the local data distributions and the sampling results in each round have been decided, but $GSI$ can reflect the reliability of the sampling results and be further used to mitigate the negative impacts of bad sampling results. Specifically, if the training data is non-i.i.d., and if the sampled clients have more similar local data distributions, which will be reflected in more similar local gradients and smaller $GSI$, the merged data distribution of sampled clients is more likely to be skewed from the global data distribution. Then the aggregated server gradient may not be fully trusted, and the server learning rate should be relatively smaller. Otherwise, the $GSI$ is larger, and the server learning rate should be larger.

### 3.3 GRADIENT SIMILARITY–AWARE LEARNING RATE ADAPTATION FOR FEDERATED LEARNING

Though we can not calculate the exact $\eta_o^t$ from Eq. (8), the above analysis motivates us to use $GSI$ as an indicator to relatively adjust the current round's server learning rate. Specifically, we can relatively adjust the learning rate by comparing the current round's $GSI^t$ with a baseline $B^t$, which we choose it as the exponential average of historical $GSI$s:

$$B^t = \beta B^{t-1} + (1 - \beta)GSI^{t-1}, \quad t = 1, 2, \cdots, \quad B^0 = GSI^0. \quad (10)$$

Then the adjusted learning rate $\hat{\eta}_t$ is calculated by multiplying the initial global learning rate $\eta_0$[5] with the ratio of $GSI^t$ and $B^t$:

$$\hat{\eta}^t = \eta_0 \left( GSI^t/B^t \right). \quad (11)$$

**Parameter group–wise adaptation** Based on the above analysis, a direct solution is to regard all parameters as a whole and universally adjust the global learning rate based on Eq. (11). However,

---

[5]$\eta_0$ is a hyper-parameter that can be tuned in the same way as previous methods did to get their optimal server learning rate $\eta_s$. Thus, this is not an extra hyper-parameter we introduced for our work.

in our preliminary explorations, we visualize the fluctuation patterns of $GSI$ in different parameter groups (i.e., weight matrices) when using FedAvg to train on 20NewsGroups (Lang, 1995)[6] in Figure 3, and find that the $GSI$s have very different fluctuation degrees in different parameter groups whether from the same layer or not. Thus, to achieve more precise adaptations, we give each parameter group $\boldsymbol{\theta}_P$ an independent adaptive learning rate based on its own $GSI$ in each round:

$$\hat{\eta}_P^t = \eta_0 \left(GSI_P^t/B_P^t\right), \quad t = 0, 1, \cdots, \tag{12}$$

where $GSI_P^t$ and $B_P^t$ are calculated based on the averaged gradient of the $\boldsymbol{\theta}_P^t$ only from Eq. (9).

**Adaptation with dynamic bounds**   Moreover, as we can see from Figure 3, the $GSI$ varies much more greatly at the beginning than it does when the training becomes stable. The reason is that, when the global model is newly initialized, the local gradients are not stable, leading to the great variability of $GSI$s. In addition, there are not enough historical $GSI$s for estimating $B_P^t$ at the beginning, which will further cause the $GSI_P^t/B_P^t$ and $\hat{\eta}_P^t$ to vary greatly. Therefore, to avoid doing harm to the initial training caused by the great variations of the learning rates, we set dynamic bounds which will restrict the parameter group–wise adaptive learning rates at the initial stage but gradually relax restrictions as the training becomes stable. The finally used learning rate for $\boldsymbol{\theta}_P^t$ is:

$$\tilde{\eta}_P^t = \eta_0 \min\{\max\{\hat{\eta}_P^t/\eta_0, 1 - \gamma t\}, 1 + \gamma t\}, \quad t = 0, 1, \cdots, \tag{13}$$

where $\gamma$ is the bounding factor that controls the steepness of dynamic bounds. We name our method the ***G**radient Similarity–Aware **L**earning RAte AD**a**ptation for **Fed**erated learning (**FedGLAD**)*. The simplified illustration is in Algorithm 1, and the detailed version is in Appendix F.

Though the previous analysis is based on the FedAvg framework, our proposed algorithm can also be applied with other federated optimization methods (e.g., FedProx (Li et al., 2018), FedAvgM (Hsu et al., 2019), FedAdam (Reddi et al., 2020)). We make a detailed discussion about these variations, and illustrate how to effectively combine FedGLAD with them in Appendix F. Also, we have a discussion about our method's limitations in the Limitations Section in Appendix A.

## 4 EXPERIMENTS AND ANALYSIS

### 4.1 DATASETS, MODELS, DATA PARTITIONING AND CLIENT SAMPLING

We perform experiments on two image classification tasks: CIFAR-10 (Krizhevsky et al., 2009) and MNIST (LeCun et al., 1998), and two text classification tasks: 20NewsGroups (Lang, 1995) and AGNews (Zhang et al., 2015). The statistics of all datasets can be found in Appendix G. We use the ResNet-56 (He et al., 2016) as the global model for CIFAR-10 and the same convolutional neural network (CNN) as that used in Reddi et al. (2020) for MNIST. We use DistilBERT (Sanh et al., 2019) as the backbone for two text classification datasets following recent studies (Hilmkil et al., 2021; Lin et al., 2021). We also explore the performance of using BiLSTM (Hochreiter & Schmidhuber, 1997) as the backbone in Appendix O.

To simulate realistic non-i.i.d. data partitions, we follow previous studies (Wang et al., 2019a; Lin et al., 2021) by taking the advantage of the Dirichlet distribution $\text{Dir}(\alpha)$. In our main experiments, we choose the non-i.i.d. degree hyper-parameter $\alpha$ as 0.1. We also conduct extra experiments on $\alpha = 1.0$. The results are in Appendix L, and the conclusions remain the same.

The total number of clients for each dataset is all 100, and the number of clients sampled in each round is 10. We adopt the widely-used *uniform sampling without replacement* strategy for client sampling (McMahan et al., 2017; Fraboni et al., 2021b). We further explore the impacts of using different sampling ratios and sampling strategies in Section 5.2 and Section 5.3 respectively.

### 4.2 BASELINE METHODS AND TRAINING DETAILS

We choose 4 popular federated optimization methods as baselines,[7] which can be divided as: (1) **Using SGD as the server optimizer:** FedAvg (McMahan et al., 2017) and FedProx (Li et al.,

---

[6]The experimental settings are the same as that introduced in Section 4.1. We also visualize the results on CIFAR-10 (Krizhevsky et al., 2009) and put the results in Appendix D, while the conclusions are the same.

[7]We also perform experiments with SCAFFOLD (Karimireddy et al., 2020), but we find it can not work well in our main experiments' settings. The detailed discussions can be found in Appendix N.

Table 1: The main results of all baselines with (+FedGLAD) or without our method.

| Method | CIFAR-10 | MNIST | 20NewsGroups | AGNews |
|---|---|---|---|---|
| | SGD Server Optimizer | | | |
| FedAvg | 35.30 ($\pm$ 1.62) | 78.17 ($\pm$ 0.35) | 72.84 ($\pm$ 0.12) | 79.48 ($\pm$ 0.46) |
| + FedGLAD | **38.44** ($\pm$ 0.91) | **79.71** ($\pm$ 0.49) | **73.14** ($\pm$ 0.08) | **80.74** ($\pm$ 0.79) |
| FedProx | 43.48 ($\pm$ 0.79) | 82.72 ($\pm$ 0.25) | 72.78 ($\pm$ 0.20) | 82.61 ($\pm$ 0.56) |
| + FedGLAD | **44.52** ($\pm$ 0.72) | **83.54** ($\pm$ 0.34) | **73.19** ($\pm$ 0.08) | **83.44** ($\pm$ 0.44) |
| | Advanced Server Optimizer | | | |
| FedAvgM | 48.79 ($\pm$ 1.35) | 81.71 ($\pm$ 0.99) | 75.44 ($\pm$ 0.10) | 80.28 ($\pm$ 1.75) |
| + FedGLAD | **50.91** ($\pm$ 1.33) | **82.11** ($\pm$ 0.64) | **75.91** ($\pm$ 0.29) | **83.68** ($\pm$ 0.87) |
| FedAdam | 56.82 ($\pm$ 1.12) | 96.94 ($\pm$ 0.14) | **76.76** ($\pm$ 1.17) | 86.72 ($\pm$ 1.02) |
| + FedGLAD | **57.69** ($\pm$ 1.23) | **97.11** ($\pm$ 0.11) | 76.59 ($\pm$ 0.81) | **88.47** ($\pm$ 0.51) |

2018); (2) **Using advanced server optimizers:** FedAvgM (Hsu et al., 2019; Wang et al., 2019b) and FedAdam (Reddi et al., 2020). The detailed descriptions of them are in the Appendix I. We show that our method can be applied to any of them to further improve the performance.

As for the local training, we choose the client optimizers for image and text tasks as SGDM and Adam respectively. For each dataset, we adopt the optimal local learning rate $\eta_l$ and batch size $B$ which are optimal in the centralized training as the local training hyper-parameters for all federated baselines. Then we allow each baseline method to tune its own best server learning rate $\eta_s$ on each dataset. The number of local epochs is 5 for two image classification datasets, and 1 for two text classification datasets. We also conduct the experiments to explore the impact of the number of local epochs $E$, the results and analysis are in Appendix P. The optimal training hyper-parameters (e.g., server learning rates) will be tuned based on the training loss of the previous 20% rounds (Reddi et al., 2020). Details of all the training hyper-parameters (including the total number of communication rounds for each dataset) and further explanations are in Appendix J.

When applying our method with each baseline method in each setting, **we use the original training hyper-parameters tuned before in that experiment directly without extra tuning**. In Appendix M, we explore the sensitivity to the bounding factor $\gamma$ of our method, and find $\gamma = 0.02$ can be a generally good choice in all cases. Thus, in our main experiments, we fix $\gamma$ as 0.02. The exponential decay factor $\beta$ is fixed as 0.9. We run each experiment on 3 different seeds, and on each seed we save the average test accuracy (%) over the last 10 rounds (Reddi et al., 2020). Finally, we report the mean and the standard deviation of the average accuracy over the last 10 rounds on all seeds. Our code is mainly based on FedNLP (Lin et al., 2021) and FedML (He et al., 2020).

### 4.3 MAIN RESULTS

The results of our main experiments are displayed in Table 1. We also display the results of the centralized training on each dataset in Appendix K. The main conclusion is, **our method can consistently bring improvement to all baselines on both image and text classification tasks in almost all settings**. Note that though the standard deviation in each experiment in Table 1 may seems a bit large compared with the improvement gap, we point out that actually **on almost each seed of one specific experiment, applying our method can gain the improvement over the corresponding baseline.** We put the result on each seed in Appendix Q for comparison.

(1) First of all, our method effectively improves the performance of FedAvg in all settings, which directly validates our analysis in Section 3.3 that it is better to set adaptive server learning rates based on the client sampling results. FedProx also uses SGD as the server optimizer but tries to improve the local gradients by making them have more consistent directions. However, FedProx requires the carefully tuning of its hyper-parameter $\mu$ to achieve better performance than FedAvg (refer to Appendix J.4), and there is no guarantee that FedProx can outperform FedAvg in any setting (same conclusion for SCAFFOLD). This means that empirically adjusting the directions of local gradients may bring negative effects. Additionally, we find that FedAvg applied with FedGLAD can achieve

comparable performance with FedProx in some cases. This result indicates that **optimizing the scales of the averaged gradients by adjusting the server learning rates is equally as important as optimizing the directions of local gradients before aggregation**, while the former is overlooked by previous studies. Also, our method can further help FedProx to gain more improvement, since they are two complementary aspects for optimizing the full target in Eq. (6).

(2) FedAvgM and FedAdam belong to another type of baselines, which uses advanced server optimizers. For these two methods, the averaged server gradient in the current round will continuously affect the training in the future rounds since there is a momentum term to store it. Our method aims to enlarge the scales of the relatively reliable averaged gradients, while down-scaling the averaged gradients that deviate greatly from the global optimization direction in specific rounds. Thus, **our method can improve the learning when server optimizers include a momentum term to stabilize training**.

(3) When comparing the results in Table 1 with the results on $\alpha = 1.0$ in Table 6 in Appendix L, we can get that on each dataset, the trend is **the improvement brought by our method is greater if the non-i.i.d. degree is larger (i.e., smaller $\alpha$)**. Our explanation is when the label distribution is more skewed, the impact of client sampling becomes larger. More analysis is in Appendix L.

### 4.4 THE EFFECT OF FEDGLAD UNDER EXTREMELY BAD CLIENT SAMPLING SCENARIOS

In Figure 1, we draw an example under the extremely bad client sampling scenario to illustrate the necessity of adapting server learning rates according to the client sampling results. Here, we validate this motivation by conducting a special experiment on MNIST with FedAvg. Specifically, while keeping other settings unchanged, we make sure that clients sampled in the 40-th round only have almost one same category of samples. Figure 4 displays the curves of the evaluation accuracy and the $GSI$ calculated by regarding all parameters as a whole after each round.

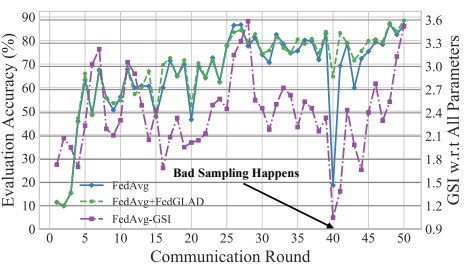

Figure 4: The good effect of FedGLAD on stabilizing training when extremely bad client sampling scenarios happen.

We notice that in the 40-th round, the accuracy of FedAvg drops greatly due to the bad client sampling results, and the corresponding $GSI$ is also the smallest (i.e., close to 1) during the training. After applying FedGLAD, the accuracy degradation problem is effectively mitigated, and the accuracy curve is much more stable. The indicates that, **our method can help to stabilize the training when extremely bad client sampling scenarios happen**.

## 5 DEEP EXPLORATIONS

In this section, we take steps further to make more explorations of our method. We choose FedAvg and FedAvgM as baseline methods, and perform experiments on CIFAR-10 and 20NewsGroups (with the backbone DistilBERT) datasets with the non-i.i.d. degree $\alpha = 0.1$. The bounding factor $\gamma$ of our method is fixed as $0.02$. All experiments are run on 3 random seeds.

### 5.1 THE EFFECT OF ADJUSTING LEARNING RATES INDIVIDUALLY FOR DIFFERENT PARAMETER GROUPS

Here, we explore the benefits of using parameter group–wise adaptation compared with the direct solution that considers all parameters as a whole and adjusts the server learning rate universally (**universal adaptation**, mentioned at the beginning of Section 3.3). The results are in Figure 5.

Firstly, the universal adaptation strategy can already bring improvement to the baselines without adaptation. This indicates that dynamically adjusting the server learning rates based on the dissimilarity between local gradients can indeed help federated learning. Moreover, the performance of parameter group–wise adaptation strategy is consistently better than the universal adaptation strategy, which validates our motivation based on the findings in Figure 3 that the non-i.i.d. data distribution has different impacts on the different parameter groups, so we should deal with them individually.

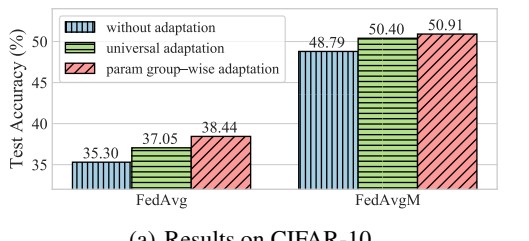
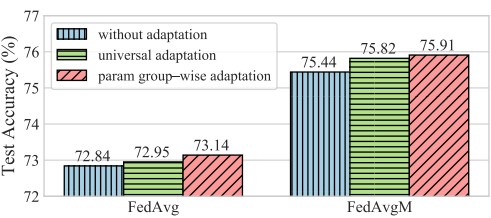

(a) Results on CIFAR-10.

(b) Results on 20NewsGroups.

Figure 5: Comparisons between two learning rate adaptation strategies of FedGLAD.

Table 2: Results under different sampling ratios.   Table 3: Results of different sampling strategies.

| Dataset | Method | Sampling Ratio | | |
|---|---|---|---|---|
| | | 5% | 10% | 20% |
| CIFAR-10 | FedAvg | 29.91 | 35.30 | 52.05 |
| | + FedGLAD | **33.35** | **38.44** | **53.40** |
| | FedAvgM | 34.72 | 48.79 | 63.56 |
| | + FedGLAD | **36.48** | **50.91** | **63.84** |
| 20NewsGroups | FedAvg | 63.09 | 72.84 | 76.31 |
| | + FedGLAD | **63.49** | **73.14** | **76.35** |
| | FedAvgM | 61.51 | 75.44 | **80.08** |
| | + FedGLAD | **62.48** | **75.91** | 79.98 |

| Dataset | Method | Sampling Strategy | |
|---|---|---|---|
| | | MD | AdaFL |
| CIFAR-10 | FedAvg | 41.44 | 39.85 |
| | + FedGLAD | **42.20** | **41.39** |
| | FedAvgM | 51.77 | 52.76 |
| | + FedGLAD | **53.85** | **53.68** |
| 20NewsGroups | FedAvg | 73.36 | 72.14 |
| | + FedGLAD | **74.04** | **72.92** |
| | FedAvgM | 75.22 | 76.18 |
| | + FedGLAD | **75.60** | **76.40** |

## 5.2 THE IMPACT OF USING DIFFERENT SAMPLING RATIOS

In the main experiments, we fix the sampling ratio as 0.1. Here, we explore our method's performance in the settings where 5, 10, 20 out of 100 clients are sampled in each round. The optimal server learning rates for baselines are re-tuned here. The results are in Table 2. The overall trend is **the improvement brought by our method is more significant when the sampling ratio is smaller**. That is because when the number of sampled clients increases, the merged data distribution of sampled clients is more likely to be consistent with the merged data distribution of all clients, thus the impact of client sampling decreases.

## 5.3 COMBINING WITH VARIOUS CLIENT SAMPLING STRATEGIES

In this subsection, we want to show that our method can also be combined with other sampling mechanisms besides the normal one used in our main experiments. We conduct experiments with two additional sampling strategies: (1) **Multinomial Distribution–based Sampling (MD)** (Fraboni et al., 2021b): considering sampled clients as $r$ i.i.d. samples from a Multinomial Distribution; (2) **Dynamic Attention-based Sampling (AdaFL)** (Chen et al., 2021): dynamically adjusting the sampling weights of clients during the training. The sampling ratio is 0.1. The results in Table 3 show that our method can also work well with these sampling strategies, since they aim to sample better groups of clients in each round while our method improves the learning after the sampling results are determined.

## 6 CONCLUSION

In this paper, we study and mitigate the difficulty of federated learning on non-i.i.d. data caused by the client sampling practice. We point out that the negative client sampling will make the aggregated gradients unreliable, thus the server learning rates should be dynamically adjusted according to the situations of the local data distributions of sampled clients. We then theoretically find that the optimal server learning rate in each round is positively related to an indicator, which can reflect the merged data distribution of sampled clients in that round. Based on this indicator, we design a novel learning rate adaptation mechanism to mitigate the negative impact of client sampling. Extensive experiments validate the effectiveness of our method.

## ETHICS STATEMENT

Though the purpose of our work is positive as we aim to improve the performance of improve federated learning on non-i.i.d. data, we admit that some adversarial clients may utilize the characteristics of our method to enforce the server to focus on their outlier local gradients, since our method tends to give larger server learning rates when local gradients have more divergent directions. However, safety problems are not in the scope of this paper and belong to another field. One possible solution is applying existing detecting techniques (Blanchard et al., 2017) to filter out extremely abnormal gradients first before aggregation.

## REPRODUCIBILITY STATEMENT

We have made great efforts to ensure the reproducibility of our work. First, we provide the necessary descriptions about the experimental settings in Section 4.1 and Section 4.2, such as describing the datasets, backbone models and baseline methods we use, and the sufficient information of the training details. Also, as we mentioned and referred in above sections, we put the complete experimental details about the datasets and models in Appendix G, the data partitioning procedure in Appendix H, the descriptions of the baseline methods in Appendix I, and the detailed training procedure and hyper-parameters in Appendix J. Most importantly, we have attached the anonymized code in the supplementary materials.

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

## A    LIMITATIONS

There are also some limitations of our method: (1) First, we do not provide the convergence analysis of our algorithm. However, our algorithm's convergence can be guaranteed by the convergence results of existing federated learning algorithms (Li et al., 2019; Karimireddy et al., 2020), since our approach is directly plugged into these methods without changing their main frameworks and our adaptive learning rate is well-bounded by the common assumption that the heterogeneity of local gradients is bounded. (2) Second, in Eq. (8), we make the assumption that the term $\sqrt{\frac{1}{r}\sum_{k\in S_t}\|\boldsymbol{g}_k^t\|^2}$ is barely affected by client sampling, while this assumption only strictly holds when the local optimizer is Adam. However, we provide the empirical evidence in the following Appendix C to show that this assumption may still roughly holds even when the local optimizer is SGD(M), as long as the number of local epochs is generally small.

## B    THE IMPACT OF CLIENT SAMPLING WHEN CLIENTS PERFORM MULTIPLE LOCAL UPDATES

In our main paper, we take the example of FedSGD (McMahan et al., 2017) to illustrate our claim that, client sampling is the main root cause of the performance degradation of federated learning on non-i.i.d. data. In real cases, clients usually perform multiple local updates before uploading the accumulated gradients (McMahan et al., 2017), in order to accelerate the convergence. In this case, previous studies (Karimireddy et al., 2020) point out that even under the full client participation setting, there still exists the optimization difficulty of FedAvg. The reason is, when the number of local steps increases, the local gradients are accumulated to have more divergent directions, which leads to the unstable aggregation of local gradients in the server. However, this is not contradictory to our claim that, in this scenario, the directions of the averaged server gradients should still be more reliable and accurate under the full client participation setting than that under the partial participation setting.

## C    DISCUSSION ABOUT THE IMPACT OF CLIENT SAMPLING ON THE SCALE COMPONENT

In Eq. (8) we made the further derivation by claiming that compared with the $GSI$, the scale component $\sqrt{\frac{1}{r}\sum_{k\in S_t}\|\boldsymbol{g}_k^t\|^2}$ is barely affected by the client sampling results. We admit that this claim can hold under some certain assumptions. (1) First, if the local optimizer is chosen as Adam, the norm of each local gradient is almost the same. That is because the update scale by using Adam is independent of the scales of gradients, but only depends on the learning rate and the number of updates, which are assumed to be the same across clients. (2) Second, if SGD(M) is the local optimizer, it is true that the scales of local gradients are different across clients, especially under the non-i.i.d. data scenario. However, if the number of local epochs is small, the difference between the norms of local gradients can be negligible compared with the divergence between their directions. This assumption can hold in ours setting since previous studies (Li et al., 2018) suggest to use relatively small number of local epochs under the non-i.i.d. data situation with large number of clients and the small sampling ratio. The reason is, in this case, large local epochs may cause the local model to overfit the local data severely and make the aggregation difficult.

In order to explore the impact of client sampling on the scale component, we conduct the following experiments: we calculate the mean and the standard deviation of the squared norms of the uploaded local gradients in each round,[8] and then calculate the ratio between the standard deviation and the mean value. We report the averaged value of the ratios in each run. If the ratio is small during the training, we can have that the scale component is indeed barely affected by the client sampling.

The results are in Table 7. As we can see, (1) when the local optimizer is Adam (i.e., experiments on 20NewsGroups), the ratios are all less than 0.2 in all settings, and the ratios are smaller when $\alpha$ is smaller or the number of epochs decreases. This indicates that when Adam is the local optimizer, our assumption that the squared norms of local gradients are almost the same across clients can hold

---

[8]The optimization method is FedAvg and the settings are kept as the same as that in the main experiments.

Figure 7: The averaged ratio values between the mean and standard deviation of uploaded local gradients in different settings.

| Dataset | $\alpha$ | Local Epochs | | |
|---|---|---|---|---|
| | | 1 | 3 | 5 |
| CIFAR-10 | 0.1 | 0.24 | 0.29 | 0.33 |
| | 1.0 | 0.23 | 0.19 | 0.15 |
| 20NewsGroups | 0.1 | 0.12 | 0.15 | 0.16 |
| | 1.0 | 0.08 | 0.15 | 0.15 |

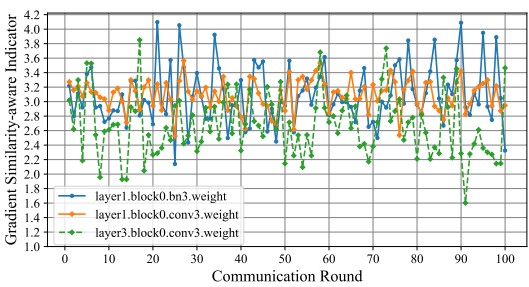

Figure 8: Fluctuation patterns of $GSI$s in different parameter groups when federated training on CIFAR-10 (Krizhevsky et al., 2009) with ResNet-56 (He et al., 2016).

well. (2) When the local optimizer is SGDM (i.e., experiments on CIFAR-10), the ratios become larger as expected. However, even when the number of local epochs is 5 and $\alpha = 0.1$, the ratio is smaller than $1/3$, so our assumption that the scale component is barely affected by the client sampling compared with the direction component still roughly holds, and the experimental results also validate the effectiveness of our method even when we choose SGDM as the local optimizer. An interesting phenomenon in CIFAR-10 is, when $\alpha = 1.0$, the ratio becomes smaller as the number of local epochs increases, but is still smaller than that when $\alpha = 0.1$. Our explanation is, when the number of local epochs is small, the randomness (e.g., data random shuffle, dropout) during the local training plays a great role; and when the number of local epochs is large enough, the scale of each local gradient is stable and only depends on the local training objective.

## D    VISUALIZATIONS OF FLUCTUATION PATTERNS OF $GSI$ ON CIFAR-10

In this section, we display the visualizations of fluctuation patterns of $GSI$ in different parameter groups (i.e., weight matrices) during federated training on CIFAR-10 (Krizhevsky et al., 2009) with ResNet-56 (He et al., 2016). Results are in Figure 8. The conclusion is the same as that in 20News-Groups (refer to Figure 3): $GSI$s have different fluctuation degrees in different kinds of parameter groups.

## E    WHY CHOOSE PARAMETER GROUP−WISE ADAPTATION

In Section 3.3, we point out that giving different adaptive learning rates to different parameter groups (i.e., weight matrices) can achieve more precise adjustments compared with the universal adaptation strategy, which considers all parameters as a whole group. We verify the benefit of using the parameter group−wise adaptation strategy in Section 5.1. Researchers can also choose other adaptation strategies, such as adjusting the learning rates layer-wisely that consider all parameters in the same layer as a group. However, we find the layer-wise adaptation performs worse than the parameter group−wise adaptation. The reason is, even in the same layer, $GSI$ has different fluctuation patterns in different kinds of weight matrices (refer to Figure 3 and Figure 8). Thus, the parameter group−wise adaptation strategy should be better. On the other hand, we point out that an individual group to be adjusted should contain a certain number of parameters, in order to make the $GSI$ more reliable due to the fact that the neurons in the same global model that are dropped out during different clients' local training are different. This indicates that the element-wise adaptation (i.e., each element in a weight matrix is re-scaled independently based on its own $GSI$) may not work well.

## F    COMBINING FEDGLAD WITH VARIOUS FEDERATED OPTIMIZATION METHODS

In our main paper, we put a simplified version of our method when combining it with FedAvg in Algorithm 1. Here, we will give the detailed versions of our method when combined with various

optimization frameworks. In the following, we will first illustrate how to apply our method in different kinds of methods.

If the optimization method aims to improve the local training while following the same aggregation mechanism as that of FedAvg, such as FedProx (Li et al., 2018), SCAFFOLD (Karimireddy et al., 2020) or FedDyn (Acar et al., 2020), we can adapt the server learning rates following the same way as that in Algorithm 1.

There are some other methods (Wang et al., 2019b; Hsu et al., 2019) which try to utilize a momentum term of the server gradient to help aggregating. In this case, we reformulate the original optimization target in Eq. (5) as:

$$
\eta_o^t = \arg\min_{\eta^t} \|(\eta^t \boldsymbol{g}^t + \beta_m \boldsymbol{m}^t) - (\boldsymbol{g}_c^t + \beta_m \boldsymbol{m}^t)\|^2 = \arg\min_{\eta^t} \|\eta^t \boldsymbol{g}^t - \boldsymbol{g}_c^t\|^2,
$$
$$
\boldsymbol{g}^t = \frac{1}{r} \sum_{k \in S^t} \boldsymbol{g}_k^t, \quad \boldsymbol{m}^t = \eta_o^{t-1} \boldsymbol{g}^{t-1} + \boldsymbol{m}^{t-1}.
$$

(14)

Thus, the solution can be written as

$$
\eta_o^t = \frac{<\boldsymbol{g}^t, \boldsymbol{g}_c^t>}{\|\boldsymbol{g}^t\|^2},
$$

(15)

which is the same as that in Eq. (7) with $\eta_s = 1.0$. Therefore, in this case, we can first adjust the current aggregated gradient's scale following the same mechanism discussed through Section 3.3. Then we use the adjusted server gradient to update the momentum, and update the global model with the global learning rate.

If the federated optimization framework (Reddi et al., 2020) utilizes an adaptive optimizer such as Adam (Kingma & Ba, 2014) or AdaGrad (Duchi et al., 2011) to update the global model in the server, our solution is to only adjust the scale of the server gradient used in the numerator while keeping the squared gradient in the denominator unchanged. There are two reasons: (1) The scale of the server gradient will not only affect the first moment estimate $\boldsymbol{m}^t$ in the numerator, but also the second moment estimate $\boldsymbol{v}^t$ in the denominator. For example, if we decrease the scale of current server gradient, then if the squared gradient $\boldsymbol{g}^t \odot \boldsymbol{g}^t$ is also correspondingly down-scaled, the second moment estimate $\boldsymbol{v}^t$ in the denominator will also decrease, which may lead to larger adaptive learning rate, and this is not we expect to happen. (2) The second reason is, the purpose of using the second moment estimate in Adam is to estimate and adjust the scale of current gradient based on the previous steps' gradients as the training goes on, while the purpose of our method is to optimize the scale of current gradient to mitigate the negative effect brought by the randomness of client sampling at a given round. Thus, we choose to keep the scale of the squared gradient unchanged.

We summarize all above variations in detail in Algorithm 1.

## G   DATASETS AND MODELS

Here we provide the detailed descriptions of the datasets and models we used in our experiments in Table 4. We get data from their public releases. Since these datasets are widely used, they do not contain any personally private information. Some of them do not have the public licenses, so we give all the public links to the datasets we use in the following. Following previous studies (Reddi et al., 2020), we use all original training samples for federated training and use test examples for final evaluation. We tune the hyper-parameters based on the averaged training loss of the previous 20% communication rounds (Reddi et al., 2020).

We use the ResNet-56 (He et al., 2016) as the global model for CIFAR-10[9] (Krizhevsky et al., 2009), and the same CNN used in Reddi et al. (2020) for MNIST[10] (LeCun et al., 1998). For 20News-Groups[11] (Lang, 1995) and AGNews[12] (Zhang et al., 2015), we choose the DistilBERT (Sanh et al., 2019) as the global model, since previous work (Hilmkil et al., 2021) validates the great effectiveness of utilizing pre-trained language models for federated fine-tuning, and DistilBERT is a

---

[9]CIFAR-10 can be obtained from https://www.cs.toronto.edu/ kriz/cifar.html.

[10]MNIST can be obtained from http://yann.lecun.com/exdb/mnist/.

[11]20NewsGroups can be obtained from http://qwone.com/ jason/20Newsgroups/.

[12]The source page of AGNews corpus is http://groups.di.unipi.it/ gulli/AG_corpus_of_news_articles.html.

---

**Algorithm 1** Variations of Gradient Similarity–Aware Learning Rate Adaptation for Federated Learning

---

1: **Server Input:** initial global model $\boldsymbol{\theta}^0$, global learning rate $\eta_0$, hyper-parameters $\beta = 0.9$, $\gamma = 0.02$ and $\beta_1, \beta_2$
2: **for** each round $t = 0$ **to** $T$ **do**
3:     Sample $r$ clients from $N$ clients: $S^t \subset \{1, \cdots, N\}$
4:     **for** $k \in S^t$ **in parallel do**
5:         Broadcast current global model $\boldsymbol{\theta}^t$ to client $k$
6:         $\boldsymbol{g}_k^t \leftarrow$ **LocalTraining**$(k, \boldsymbol{\theta}^t, D_k)$
7:     **end for**
8:     $\boldsymbol{g}^t \leftarrow \frac{1}{r} \sum_{k \in S^t} \boldsymbol{g}_k^t$
9:     $\boldsymbol{v}^{t+1} \leftarrow \beta_2 \boldsymbol{v}^t + (1 - \beta_2) \boldsymbol{g}^t \odot \boldsymbol{g}^t$ (FedAdam)
10:     **for** each parameter group $\boldsymbol{\theta}_P^t$'s gradient $\boldsymbol{g}_p^t$ **do**
11:         $GSI_P^t \leftarrow \sqrt{(\sum_{k \in S_t} \|\boldsymbol{g}_{P,k}^t\|^2) / (r \|\boldsymbol{g}_P^t\|^2)}$
12:         **if** $t = 0$ **then**
13:             $B_P^t \leftarrow GSI_P^t$
14:         **end if**
15:         $\tilde{\eta}_P^t \leftarrow \min\{\max\{GSI_P^t / B_P^t, 1 - \gamma t\}, 1 + \gamma t\}$
16:         $B_P^{t+1} \leftarrow \beta B_P^t + (1 - \beta) GSI_P^t$
17:         $\boldsymbol{g}_P^t \leftarrow \tilde{\eta}_P^t \boldsymbol{g}_P^t$
18:     **end for**
19:     $\boldsymbol{\theta}^{t+1} \leftarrow \boldsymbol{\theta}^t - \eta_0 \boldsymbol{g}^t$ (FedAvg, FedProx, SCAFFOLD)
20:     $\boldsymbol{m}^{t+1} \leftarrow \beta_1 \boldsymbol{m}^t + (1 - \beta_1) \boldsymbol{g}^t$
21:     $\boldsymbol{\theta}^{t+1} \leftarrow$ **SGDM**$(\boldsymbol{\theta}^t, \boldsymbol{m}^{t+1}, \eta_0)$ (FedAvgM)
22:     $\boldsymbol{\theta}^{t+1} \leftarrow$ **Adam**$(\boldsymbol{\theta}^t, \boldsymbol{m}^{t+1}, \boldsymbol{v}^{t+1}, \eta_0)$ (FedAdam)
23: **end for**
24:
25: **Client $k$'s Input:** current global model $\theta^t$, local dataset $D_k$, local training hyper-parameters
26: **LocalTraining**$(k, \boldsymbol{\theta}^t, D_k)$:
27: Client $k$ performs centralized training to $\boldsymbol{\theta}^t$ on local data $D_k$ with tuned training hyper-parameters, get final model $\boldsymbol{\theta}_k^{t+1}$
28: **Return** $\boldsymbol{g}_k^t = \boldsymbol{\theta}^t - \boldsymbol{\theta}_k^{t+1}$

---

Table 4: The statistics of the datasets we used in our experiments.

| Dataset | # of Training Samples | # of Testing Samples | # of Labels | Model | # of Rounds |
|---|---|---|---|---|---|
| CIFAR-10 | 50,000 | 10,000 | 10 | ResNet-56 | 500 |
| MNIST | 60,000 | 10,000 | 10 | CNN | 50 |
| 20NewsGroups | 11,314 | 7,532 | 20 | DistilBERT | 100 |
| AGNews | 120,000 | 7,600 | 4 | DistilBERT | 100 |

powerful lite version of BERT (Devlin et al., 2019) which makes it more suitable to local devices. The `max_seq_length` is set as 128. We also conduct experiments on BiLSTM (Hochreiter & Schmidhuber, 1997), and display the results in Appendix O.

## H    DATA PARTITIONING STRATEGY

To simulate the non-i.i.d. data partitions, we follow FedNLP (Lin et al., 2021) by taking the advantage of the Dirichlet distribution. Specifically, assume there are totally $C$ classes in the dataset, the vector $\mathbf{q_k}$ ($q_{k,i} \geq 0, i \in [1, C], \|\mathbf{q_k}\|_1 = 1$) represents the label distribution in client $k$, then we draw $\mathbf{q_k}$ from a Dirichlet distribution $\mathbf{q_k} \sim \text{Dir}(\alpha \mathbf{p})$ where $\mathbf{p}$ represents the label distribution in the whole dataset and $\alpha$ controls the degree of skewness of label distribution: when $\alpha$ becomes smaller, the label distribution becomes more skewed; when $\alpha \to \infty$, it becomes the i.i.d. setting. We assume

each client has the same number of local training samples,[13] so we can allocate samples to client $k$ once $\mathbf{q_k}$ is determined. In order to solve the problem that some clients may not have enough examples to sample from specific classes as the sampling procedure goes, FedNLP provides a *dynamic reassigning strategy* that can use samples from other classes which still have remainings to fill the vacancy of the chosen class whose samples are all used out. More details can be found in Lin et al. (2021).

## I    BASELINE METHODS

We choose 4 widely-used federated optimization methods as baselines in our main experiments, and they are: (1) FedAvg (McMahan et al., 2017): the most popular baseline in federated learning; (2) FedProx (Li et al., 2018): the widely-adopted baseline which aims to optimize the directions of local gradients, while keeping as the same aggregation mechanism as that of FedAvg in the server; (3) FedAvgM (Hsu et al., 2019; Wang et al., 2019b): the method that adds the momentum term during the server's updates; (4) FedAdam (Reddi et al., 2020): the method that utilizes the adaptive optimizer Adam (Kingma & Ba, 2014) to update the global model with the averaged server gradient. We also conduct experiments on another baseline SCAFFOLD (Karimireddy et al., 2020), while during experiments we find SCAFFOLD itself can not converge well in our main experiments' settings. Thus, we make a separate discussion about this part in Appendix N.

## J    TRAINING DETAILS

### J.1    LOCAL TRAINING

Firstly, we find SGDM is more suitable to train CNNs and Adam is more suitable to train Transformer-based models. Thus we choose the client optimizer as SGDM when training on two image classification datasets, and choose the client optimizer as Adam for two text classification datasets. Then we perform centralized training on each dataset in order to find the optimal client learning rate $\eta_l$ and batch size $B$, and use that learning rate and batch size as the local training hyper-parameters for all federated optimization baselines in the federated training setting. The optimal local training hyper-parameters for each dataset are listed in Table 13. To decide the number of local training epoches for each dataset, we tune it from $\{1, 3, 5, 10\}$ by using FedAvg, and use the tuned one for other optimization baselines.

### J.2    SERVER AGGREGATING

We allow each baseline to tune its own optimal server learning rate in each data partition setting. The server learning rate search grids for all baseline methods are displayed in Table 14. We also display the server learning rates we have tuned and used in our main experiments in Table 15. The server learning rates used for the baseline methods in Section 5.2 and Section 5.3 are re-tuned, since we find the optimal server learning rates under different sampling ratios or different sampling strategies are different.

There are several reasons to use the above hyper-parameters' tuning strategy: (1) Since clients have local data, the optimal hyper-parameters in the centralized training setting should also work well for clients' local training in the federated training. (2) Using the same local training hyper-parameters for each baseline method allows us to make a fair and clear comparison between them.

### J.3    INFRASTRUCTURE AND TRAINING COSTS

All experiments are conducted on 4∗TITAN RTX. The running time of each method on CIFAR-10, MNIST, 20NewsGroups and AGNews is about 12h, 0.5h, 6h and 6h respectively.

---

[13]In this paper, we mainly focus on the label distribution shift problem, rather than the quantity shift problem. Also, since we assume each client performs the same number of local updates following previous studies, this practice is reasonable.

Table 5: The mean value and the standard deviation of the test accuracy for each dataset in the traditional centralized training setting.

| CIFAR-10 | MNIST | 20NewsGroups | AGNews |
|---|---|---|---|
| 91.86 ($\pm$ 0.49) | 99.26 ($\pm$ 0.04) | 84.12 ($\pm$ 0.05) | 93.43 ($\pm$ 0.07) |

### J.4 HYPER-PARAMETER'S TUNING OF FEDPROX

For FedProx, we allow it to tune its own hyper-parameter $\mu$ from $\{1.0, 0.1, 0.01, 0.001\}$ in each data partition setting, and we display the best $\mu$ in each setting in Table 12. As we can see, there is no general $\mu$ for all settings, and researchers may carefully tune this hyper-parameter in each setting.

### J.5 HYPER-PARAMETER'S TUNING OF FEDGLAD

As for our method, the key hyper-parameter is the bounding factor $\gamma$. Though the best $\gamma$ in each setting depends on its dataset, the label skewness degree and the backbone model, in our main experiments we show that there exists a general value (i.e., $\gamma = 0.02$) which works well in most settings. If researchers want to tune the optimal $\gamma$ in each setting to achieve better performance, the recommended grid is $\{0.01, 0.02, \cdots, 0.05\}$.

Since our method aims to adjust the server learning rate relatively, it is also important to choose a proper global learning rate $\eta_0$ (refer to Algorithm 1).[14] During our experiments, when applying our method, we directly use the server learning rate used in the original baseline without extra tuning, since we assume the optimal learning rate for the original baseline is the right choice for the global learning rate after applying our method, and we want to show our method is easy to apply. However, it is expectable that re-tuning the global learning rate for our method may bring more improvement.

## K RESULTS OF CENTRALIZED TRAINING

For comparison with the results in the federated training setting, we display the results on all datasets in the traditional centralized training setting in Table 5. The training settings and hyper-parameters on each dataset are kept as the same as that in the local training of the federated learning.

## L EXPERIMENTS ON $\alpha = 1.0$

Besides the non-i.i.d. degrees we set in our main paper to conduct the main experiments, we also explore the performance of our method when $\alpha = 1.0$. The baselines and the settings are the same as that in the main experiments, and the results are in Table 6.

As we can see, FedGLAD can also consistently bring improvement in this setting, which validates the effectiveness of our method. However, compared with the results in Table 1 in the main paper, we find that on each dataset, the overall trend is **the improvement brought by our method are greater if the non-i.i.d. degree is larger (i.e., smaller $\alpha$).** Our explanation is when the label distribution is more skewed, the impact of client sampling becomes larger. For example, when data is more non-i.i.d. and the sampled clients have the similar but skewed local data distributions, the averaged server gradient's direction will deviate far more away from the ideal direction that we assume is averaged by all clients' local gradients. Thus, it becomes more important to adjust the server learning rates in these scenarios.

## M THE SENSITIVITY TO THE BOUNDING FACTOR $\gamma$

Here, we want to explore our method's sensitivity to the bounding factor $\gamma$. The results are displayed in Figure 8. Generally speaking, the best $\gamma$ in each setting depends on many factors, such as the

---

[14]This can be an important hyper-parameter if the optimization method follows the same aggregation strategy as that of FedAvg. However, if using FedAvgM or FedAdam, $\eta_0 = 1.0$.

Table 6: The results of all baselines with (+FedGLAD) or without our method on $\alpha = 1.0$.

| Method | CIFAR-10 | MNIST | 20NewsGroups | AGNews |
|---|---|---|---|---|
| SGD Server Optimizer | | | | |
| FedAvg | 72.09 (± 1.27) | 91.42 (± 0.11) | 76.66 (± 0.36) | 91.27 (± 0.04) |
| + FedGLAD | **72.79** (± 1.13) | **91.63** (± 0.08) | **77.21** (± 0.42) | **91.38** (± 0.04) |
| FedProx | 71.22 (± 0.83) | 91.68 (± 0.05) | 76.51 (± 0.47) | 91.24 (± 0.09) |
| + FedGLAD | **71.52** (± 0.61) | **91.86** (± 0.06) | **77.13** (± 0.17) | **91.35** (± 0.09) |
| Advanced Server Optimizer | | | | |
| FedAvgM | 72.63 (± 0.56) | 93.37 (± 0.29) | 79.51 (± 0.31) | 90.07 (± 0.27) |
| + FedGLAD | **73.07** (± 0.33) | **93.97** (± 0.26) | **79.77** (± 0.08) | **90.45** (± 0.42) |
| FedAdam | 79.97 (± 1.30) | 98.42 (± 0.08) | 79.96 (± 0.12) | 91.72 (± 0.12) |
| + FedGLAD | **80.75** (± 1.35) | **98.53** (± 0.04) | **80.13** (± 0.20) | **91.81** (± 0.10) |

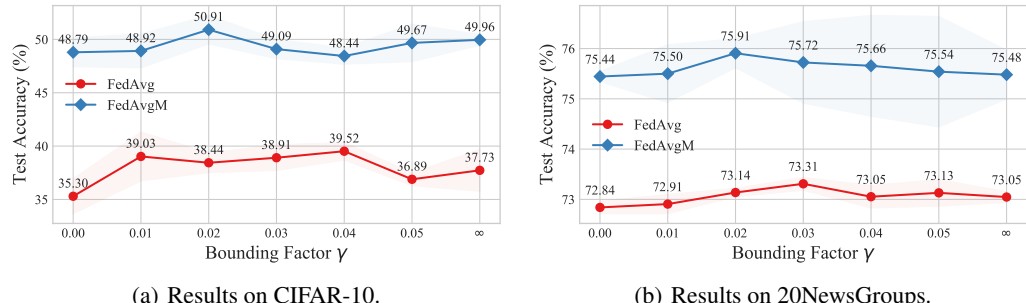

(a) Results on CIFAR-10.        (b) Results on 20NewsGroups.

Figure 8: The results of using different $\gamma$ for FedGLAD. $\gamma = 0$ represents the original baseline and $\gamma = \infty$ represents removing the bounds from the beginning. $\gamma = 0.02$ can be a generally good choice.

dataset and the non-i.i.d. degree of data distribution across clients, but from figures, we can see that there exists a general value (i.e., $\gamma = 0.02$) which works well in most cases. Therefore, we choose 0.02 as the default value in the experiments in the main paper. Moreover, (1) though the performance of removing the bounds is better than the original baseline in some cases, it is consistently worse than that of setting the bounds. (2) The standard deviation in each experiment of using larger $\gamma$ increases. This validates our motivation that without the restrictions, the learning rates at the beginning will vary greatly, which will cause harm to the training.

## N  EXPERIMENTS ON SCAFFOLD

Besides the baselines we choose in the main experiments, we also conduct experiments with another baseline SCAFFOLD (Karimireddy et al., 2020). SCAFFOLD aims to optimize the directions of the averaged server gradients by improving the local training with the help of local control variates. The original version of SCAFFOLD assumes the client optimizer is SGD, while we make some adaptations to make it applicable when the client optimizer is SGDM or Adam. However, we find the SCAFFOLD can not converge well on two image classification tasks if we follow the same training settings in our main experiments. We find the key factor which will affect the performance of SCAFFOLD is the sampling ratio and the number of local training epochs $E$. That is, if the sampling ratio is very small and when $E$ is very large, SCAFFOLD may not converge well. We guess the reason may be that, when the sampling ratio is small, the number between two consecutive rounds of participation of client $k$ is large. Thus, the local control variate of client $k$ in the latter round may no longer be reliable, and using this control variate to adjust the directions of local gradients for many local update steps will cause harm to the local training. While for two text

Table 7: The results of SCAFFOLD with (+FedGLAD) or without our method in different non-i.i.d. degree settings.

| Method | 20NewsGroups | | AGNews | | MNIST | |
|---|---|---|---|---|---|---|
| | $\alpha = 0.1$ | $\alpha = 1.0$ | $\alpha = 0.1$ | $\alpha = 1.0$ | $\alpha = 0.1$ | $\alpha = 1.0$ |
| SCAFFOLD | 76.22 (± 0.63) | 77.14 (± 0.39) | 77.43 (± 0.99) | 91.08 (± 0.15) | 86.02 (± 0.64) | **87.93** (± 0.30) |
| + FedGLAD | **76.88** (± 0.48) | **78.46** (± 0.30) | **81.42** (± 1.48) | **91.17** (± 0.16) | **86.06** (± 0.88) | 87.91 (± 0.29) |

Table 8: The results of BiLSTM.

| Method | $\alpha$ | 20NewsGroups | AGNews |
|---|---|---|---|
| Centralized | $\infty$ | 78.28 (± 0.94) | 91.47 (± 0.16) |
| FedAvg | 0.1 | 36.25 (± 0.46) | 69.84 (± 0.29) |
| + FedGLAD | 0.1 | **37.37** (± 0.40) | **70.64** (± 0.24) |
| FedAvgM | 0.1 | 35.54 (± 0.64) | 71.36 (± 1.29) |
| + FedGLAD | 0.1 | **35.91** (± 0.67) | **73.50** (± 1.40) |

classification tasks, we choose $E$ as 1 and also use a pre-trained backbone that already mitigates the problem caused by the non-i.i.d. data distribution and the client sampling at a certain degree, so the SCAFFOLD can work well on these two text classification datasets.

We keep the training hyper-parameters the same as that in our main experiments for two text classification tasks. After re-tuning, we choose $E = 1$ for two image classification tasks. Other settings are the same as that in our main experiments. We find SCAFFOLD itself can still not converge well on CIFAR-10, so we only report the results on the other three datasets in Table 7. The conclusions are the same as those in the main experiments: FedGLAD is also applicable with SCAFFOLD and tends to bring more improvement when non-i.i.d. degree is larger.

## O    EXPERIMENTS ON BiLSTM

In our main experiments, we use a pre-trained backbone DistilBERT for two text classification tasks. In this section, we conduct extra experiments with a not pre-trained backbone BiLSTM (Hochreiter & Schmidhuber, 1997) on 20NewsGroups and AGNews. The model is a 1-layer BiLSTM with the hidden size of 300. We use the pre-trained word embeddings `glove.6b.300d`[15] (Pennington et al., 2014) for the embedding layer. The `max_seq_length` is 128. The dropout ratio for the LSTM layer is 0.2.

We use FedAvg and FedAvgM as baselines, and the non-i.i.d. degree $\alpha = 0.1$. Other details can be found in Table 16. The results are in Table 8. As we can see, since BiLSTM is not pre-trained, the performance gap between the federated training and the centralized training is larger than that of using DistilBERT as the backbone under the same number of communication rounds. Secondly, after applying our learning rate adaptation mechanism, the performance consistently becomes better. The experimental results validate the effectiveness of our method on the BiLSTM.

## P    EXPERIMENTS WITH DIFFERENT NUMBERS OF LOCAL EPOCHS

In our main experiments, we set local epochs $E = 5$ for image classification tasks. As we discussed in Section 3.2 in the main paper and Appendix C, when the local optimizer is SGD(M) and when the number of epochs increases, our assumption about the scale component and the direction component may not generally holds. However, also as we explained, increasing local epochs will easily make the local model overfit to the local dataset severely, and the choices of local epochs in

---
[15]Downloaded from https://nlp.stanford.edu/projects/glove.

Table 9: The results of different numbers of local epochs.

| Method | CIFAR-10 | | MNIST | |
|---|---|---|---|---|
| | $E = 10$ | $E = 20$ | $E = 10$ | $E = 20$ |
| FedAvg | 52.23 (± 1.78) | 58.31 (± 1.59) | 82.71 (± 0.68) | 86.13 (± 0.40) |
| + FedGLAD | **53.81** (± 0.56) | **59.02** (± 1.13) | **84.25** (± 0.33) | **87.47** (± 0.35) |
| FedAvgM | 53.28 (± 1.86) | 56.59 (± 1.50) | **83.15** (± 2.63) | 86.20 (± 2.55) |
| + FedGLAD | **55.55** (± 0.24) | **58.35** (± 1.04) | 82.71 (± 2.64) | **86.68** (± 2.68) |

Table 10: Results on each seed under $\alpha = 0.1$.

| Dataset | Method | Random Seed | | |
|---|---|---|---|---|
| | | Seed1 | Seed2 | Seed3 |
| CIFAR-10 | FedAvg | 37.08 | 33.90 | 34.91 |
| | + FedGLAD | **39.49** | **38.02** | **37.82** |
| | FedAvgM | 47.24 | 49.38 | 49.76 |
| | + FedGLAD | **49.40** | **51.91** | **51.42** |
| MNIST | FedAvg | 78.29 | 77.77 | 78.44 |
| | + FedGLAD | **79.20** | **79.75** | **80.19** |
| | FedAvgM | 80.68 | 82.65 | 81.79 |
| | + FedGLAD | **81.68** | **82.84** | **81.81** |
| 20NewsGroups | FedAvg | 72.72 | 72.96 | 72.84 |
| | + FedGLAD | **73.13** | **73.22** | **73.06** |
| | FedAvgM | 75.39 | 75.38 | 75.56 |
| | + FedGLAD | **76.02** | **76.12** | **75.58** |
| AGNews | FedAvg | 78.95 | 79.78 | 79.72 |
| | + FedGLAD | **79.83** | **81.29** | **81.11** |
| | FedAvgM | 82.25 | 79.67 | 78.91 |
| | + FedGLAD | **84.22** | **84.14** | **82.67** |

Table 11: Results on each seed under $\alpha = 1.0$.

| Dataset | Method | Random Seed | | |
|---|---|---|---|---|
| | | Seed1 | Seed2 | Seed3 |
| CIFAR-10 | FedAvg | 73.20 | 70.70 | 72.36 |
| | + FedGLAD | **74.07** | **71.91** | **72.39** |
| | FedAvgM | 72.18 | 72.44 | 73.26 |
| | + FedGLAD | **72.70** | **73.18** | **73.32** |
| MNIST | FedAvg | 91.48 | 91.30 | 91.49 |
| | + FedGLAD | **91.70** | **91.55** | **91.65** |
| | FedAvgM | 93.55 | 93.52 | 93.04 |
| | + FedGLAD | **94.06** | **94.17** | **93.67** |
| 20NewsGroups | FedAvg | 76.57 | 76.35 | 77.06 |
| | + FedGLAD | **77.53** | **76.74** | **77.37** |
| | FedAvgM | 79.51 | 79.20 | 79.82 |
| | + FedGLAD | **79.80** | **79.68** | **79.83** |
| AGNews | FedAvg | 91.26 | 91.31 | 91.23 |
| | + FedGLAD | **91.38** | **91.41** | **91.34** |
| | FedAvgM | 90.28 | 89.77 | 90.17 |
| | + FedGLAD | **90.82** | **89.99** | **90.53** |

our experiments are also widely-adopted choices in previous studies (Reddi et al., 2020; Lin et al., 2021).

Here, we conduct further experiments to explore the impact of the large number of local epochs on FedGLAD. We perform experiments on CIFAR-10 and MNIST with $\alpha = 0.1$ and the local epochs $E = 10$ and $E = 20$. As we can see from the Table 9, FedGLAD can also bring improvement in almost all settings. Our explanation is, in this case, the negative effect of bad sampling results will be further amplified, and GSI can detect this anomaly and decrease the learning rate in time. Also, during training we find that the accuracy curves (also the loss curves) vary greatly when $E = 10$ or $E = 20$, indicates that they may not be proper choices in ours settings, and $E = 5$ can guarantee the stable training.

## Q  RESULTS ON EACH SEED OF EACH EXPERIMENT

As we mentioned in our paper in the experiment analysis section (Section 4.3), the standard deviation in each experiment in Table 1 and Table 6 may seems a bit large compared with the improvement gap. However, we display the result on each seed of our method and baselines (take FedAvg and FedAvgM as examples) in the Table 10 and Table 11, we point out that actually **on almost each seed of one specific experiment, applying our method can gain the improvement over the corresponding baseline.** The large standard deviation of the baseline results are mainly caused by the fact that the large non-i.i.d. degree makes the baseline training sensitive to the random seeds.

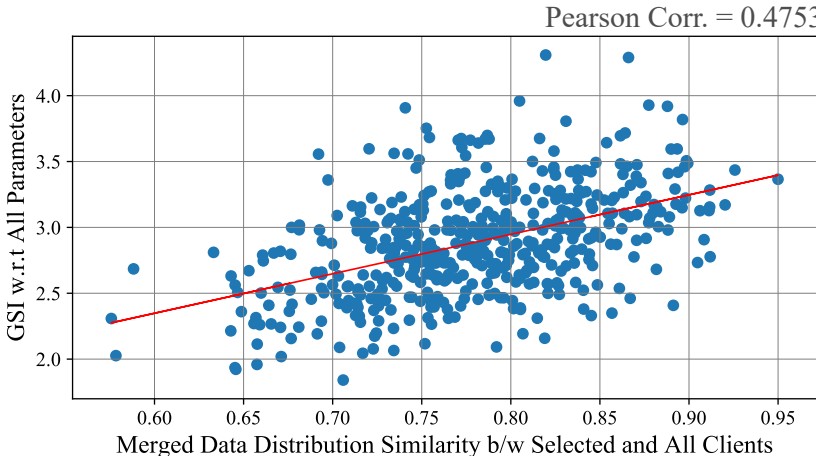

Figure 9: GSI is positively related to the the merged data distribution similarity between the sampled clients and all clients.

## R   CORRELATION BETWEEN $GSI$ AND THE MERGED DATA DISTRIBUTION SIMILARITY BETWEEN THE SAMPLED CLIENTS AND ALL CLIENTS

Here, we make deep analysis about the correlation between our proposed $GSI$ and the merged data distribution similarity between the sampled clients and all clients. First, we calculate the $GSI^t$ in each round considering all parameters as a group. At the same time, we calculate the similarity score between the merged data distribution of currently sampled clients and that of all clients. The similarity score is based on the cosine similarity between two distributions: denote $L$ as the total number of classes, $n_k^{(i)}$ as the number of samples belonging to label $i$ in client k's local data, $n_k = \sum_{i=i}^{L} n_k^i$, then the merged data distribution of sampled clients is $d^t = (\frac{\sum_{k \in S^t} n_k^{(1)}}{\sum_{k \in S^t} n_k}, \cdots, \frac{\sum_{k \in S^t} n_k^{(L)}}{\sum_{k \in S^t} n_k})$. Similarly, the merged data distribution of all clients is written as $d_c = (\frac{\sum_{k=1}^{N} n_k^{(1)}}{\sum_{k=1}^{N} n_k}, \cdots, \frac{\sum_{k=1}^{N} n_k^{(L)}}{\sum_{k=1}^{N} n_k})$. Then, the similarity score can be calculated as

$$sim\_score^t = \texttt{CosSim}(d^t, d_c).$$

Then we visualize the scatter plot of the corresponding $GSI^t$ and $sim\_score^t$ in Figure 9. The red line corresponds to the linear regression result. The Pearson correlation coefficient between two variables is 0.4753, revealing that **GSI is positively related to the the merged data distribution similarity between the sampled clients and all clients**. This indicates that the GSI can effectively reflect the sampling results, and then be used to optimize the server learning rates, which helps to verify our original motivation.

Table 12: The best $\mu$ for FedProx we tuned in each data partition setting.

| Method | CIFAR-10 | | MNIST | | 20NewsGroups | | AGNews | |
|---|---|---|---|---|---|---|---|---|
| | $\alpha = 0.1$ | $\alpha = 1.0$ | $\alpha = 0.1$ | $\alpha = 1.0$ | $\alpha = 0.1$ | $\alpha = 1.0$ | $\alpha = 0.1$ | $\alpha = 1.0$ |
| FedProx | 0.1 | 0.1 | 1.0 | 0.1 | 0.001 | 0.001 | 1.0 | 0.001 |

Table 13: Local training hyper-parameters for each dataset.

| Dataset | Learning Rate | Batch Size | Client Optimizer | Local Epochs |
|---|---|---|---|---|
| CIFAR-10 | 0.1 | 64 | SGDM | 5 |
| MNIST | 0.01 | 64 | SGDM | 5 |
| 20NewsGroups | 5e-5 | 32 | Adam | 1 |
| AGNews | 2e-5 | 32 | Adam | 1 |

Table 14: The server learning rate search grids for each baseline method.

| Methods | Datasets | Search Grids |
|---|---|---|
| {FedAvg, FedProx, SCAFFOLD} | {CIFAR-10, MNIST} | $\{0.5, 1.0, 2.0, \cdots, 10.0\}$ |
| | {20NewsGroups, AGNews} | $\{0.5, 1.0, 2.0, \cdots, 10.0\}$ |
| FedAvgM | {CIFAR-10, MNIST} | $\{0.05, 0.1, 0.2, \cdots, 0.5, 1.0, 2.0, \cdots, 5.0\}$ |
| | {20NewsGroups, AGNews} | $\{0.05, 0.1, 0.2, \cdots, 0.5, 1.0, 2.0, \cdots, 5.0\}$ |
| FedAdam | {CIFAR-10, MNIST} | $\{1e\text{-}3, 2e\text{-}3, \cdots, 5e\text{-}3, 1e\text{-}2, 2e\text{-}2, \cdots, 5e\text{-}2\}$ |
| | {20NewsGroups, AGNews} | $\{1e\text{-}5, 2e\text{-}5, \cdots, 5e\text{-}5, 1e\text{-}4, 2e\text{-}4, \cdots, 5e\text{-}5\}$ |

Table 15: The optimal server learning rates for each baseline method in our experiments.

| Method | CIFAR-10 | | MNIST | | 20NewsGroups | | AGNews | |
|---|---|---|---|---|---|---|---|---|
| | $\alpha = 0.1$ | $\alpha = 1.0$ | $\alpha = 0.1$ | $\alpha = 1.0$ | $\alpha = 0.1$ | $\alpha = 1.0$ | $\alpha = 0.1$ | $\alpha = 1.0$ |
| FedAvg | 0.5 | 0.5 | 1.0 | 1.0 | 7.0 | 6.0 | 1.0 | 1.0 |
| FedProx | 0.5 | 0.5 | 1.0 | 1.0 | 7.0 | 6.0 | 1.0 | 1.0 |
| SCAFFOLD | 0.5 | 0.5 | 1.0 | 1.0 | 7.0 | 6.0 | 1.0 | 1.0 |
| FedAvgM | 0.2 | 0.1 | 0.5 | 0.5 | 2.0 | 2.0 | 0.5 | 0.5 |
| FedAdam | 5e-3 | 5e-3 | 1e-2 | 1e-2 | 3e-4 | 3e-4 | 2e-4 | 1e-4 |

Table 16: Training details of experiments on BiLSTM.

| Dataset | Local Training | | | | Server Aggregating | | | |
|---|---|---|---|---|---|---|---|---|
| | Optimizer | LR | Batch Size | Local Epochs | Method | Server LR | Total Rounds | $\gamma$ |
| 20NewsGroups | Adam | 0.001 | 32 | 1 | FedAvg | 1.0 | 200 | 0.01 |
| | | | | | FedAvgM | 0.1 | 200 | 0.01 |
| AGNews | Adam | 0.001 | 32 | 1 | FedAvg | 0.5 | 100 | 0.02 |
| | | | | | FedAvgM | 0.05 | 100 | 0.02 |

