# OpenReview forum: "When to Trust Aggregated Gradients: Addressing Negative Client Sampling in Federated Learning"
_ICLR.cc/2023/Conference — Submitted to ICLR 2023_

### Official Review · Reviewer_VDBr · 2022-10-20

**Confidence:** 4
**Correctness:** 2
**Technical Novelty And Significance:** 3
**Empirical Novelty And Significance:** 3
**Recommendation:** 6

**Clarity, Quality, Novelty And Reproducibility:**

Clarity: The paper is well-written and easy to follow.

Quality: The algorithm improves the accuracy of federated learning model. However, it is questionable why the algorithm works.

Novelty: The proposed algorithm alters the server learning rate instead of gradient direction, which is orthogonal to previous works in federated learning. This direction has novelty in federated learning.

Reproducibility: The authors give detailed explanation of their algorithm, and provide code implementation in the supplementary material.


**Strength And Weaknesses:**

Strength:

- The intuition is clear and the paper is easy to follow.
- The paper has extensive empirical evaluation, which does show universal accuracy improvement.

Weaknesses:

- Inconsistency between intuitive claim and algorithm design. In the introduction, Figure 1, the authors claim that one should relatively decrease the server learning rate when the averaged gradient's **direction** deviates far away from the ideal gradient. However in the first inequality in Eq. (8), the authors replace the cosine similarity between $$g^t$$ and $$g_c^t$$ with 1, which totally ignores the direction information.
- Unclear meaning of GSI. At the end of page 4, the authors claim that GSI measures the normalized similarity between local gradients. However, we can consider two extreme case. (1) When all local gradients are the same, all $$g_k^t = g^t$$, then GSI is 1. (2) When client gradients are diversed, e.g., half of then is $$(2, 2)$$ while half of them are $$(2, -2)$$, the GSI will be 1.414. So when local gradients are more noisy, the aggregated gradients will be enlarged rather than decreased, which contradicts with the authors' intuition.
- The authors claim that unreliable gradients will be decreased while reliable gradients will be enlarged. There lacks verification of this important claim, e.g., a figure shows the correlation between GSI and gradient estimation error / distribution discrepancy between [merged dataset of selected clients] and [merged dataset of all clients].

**Summary Of The Paper:**

In federated learning, when clients have non-i.i.d. data, client sampling will introduce noises to the aggregated gradient. The authors propose a learning rate adaptation algorithm to adjust the server learning rate, which aims to prevent large parameter update to a bad direction. The algorithm is based on theoretical deduction, and is empirically evaluated on multiple tasks which universal accuracy improvement.


**Summary Of The Review:**

It is a novel and effective algorithm, however the explanation is not convincing.

---

> ### Author Response · Authors · 2022-11-15
> **Thank you for your questions**
>
> **Q1**: Regarding the concern about the inconsistency between intuitive claim and algorithm design.
>
> **A1**: From Eq. (7) and Eq. (8) we can see that given a specific round, the optimal server learning rate $\eta_{o}^{t}$ is mainly affected by two factors: $<g^{t}, g_{c}^{t}>/||g^{t}|| $ that measures the similarity between $g^{t}$ and $g_{c}^{t}$, and the GSI. According to our analysis, **these two factors will have similar patterns**. That is, when the sampling results are bad, $<g^{t}, g_{c}^{t}>/||g^{t}|| $ will be small and GSI will also be small (refer to the last paragraph of Section 3.2). However, since $g_{c}^{t}$ is unknown, we choose to omit the factor of  $<g^{t}, g_{c}^{t}>/||g^{t}|| $ and only use GSI as the indicator for adjusting server learning rates. This approximation will lose some information about the direction relationship between $g^{t}$ and $g_{c}^{t}$, but **it will not affect the positive correlation between the GSI and the $\eta_{o}^{t}$**.
>
> To verify that our proposed GSI can indeed be used to reflect the reliability of current sampling results, we conducted extensive experiments to the effectiveness of utilizing GSI. Furthermore, we make the analysis to explore the positive correlation between GSI and similarity between the merged data distribution of selected clients and that of all clients, please refer to **A3**.
>
> **Q2**: Regarding the question about the unclear meaning of GSI.
>
> **A2**: **The mechanism of our method exactly sets larger server learning rates when current local gradients are relatively more diverse**. Since in this case, we think the local gradients contain more comprehensive data information from diverse categories. There is a detailed discussion in the last paragraph in Section 3.2.
>
> However, as you mentioned, when the local data/gradients are noisy, our method tends to set a larger server learning rate, and we had a discussion about a similar scenario in our Ethics Statement section (at the beginning of Page 10). There are existing studies [1] managing to filter out the most confident samples (i.e., most likely the correct samples) for local training to avoid the local model overfitting on the noisy data. As they can help to denoise the local gradients during local training, **our method can be combined with these methods to deal with the federated learning with noisy labels**, since we only aim to improve the server aggregation phase.
>
>
> **Q3**: Regarding the verification of the claim that GSI can reflect the distribution discrepancy between the merged dataset of selected clients and the merged dataset of all clients.
>
> **A3**:  Thank you for your suggestion! We further make the visualizations about the relationship between the GSI and the similarity between the merged data distribution of selected clients and that of all clients, and **we put the results and analysis in Appendix R (marked in blue) in our newly updated version**. The conclusion is, **GSI is positively related to the merged data distribution similarity between the sampled clients and all clients, and can thus effectively reflect the sampling results**.
>
>
> [1] Yang, Seunghan, et al. "Robust federated learning with noisy labels." IEEE Intelligent Systems 2022

---

> > ### Comment · Reviewer_VDBr · 2022-12-04
> > **Thank you for the response!**
> >
> > I appreciate the authors' response to my comments. Some of my questions have been addressed, and I have updated my score accordingly.

---

### Official Review · Reviewer_s9jn · 2022-10-21

**Confidence:** 4
**Clarity, Quality, Novelty And Reproducibility:** The paper is clearly written and seem…
**Correctness:** 2
**Technical Novelty And Significance:** 2
**Empirical Novelty And Significance:** 2
**Recommendation:** 5

**Strength And Weaknesses:**

The paper is well written and clearly motivates the problem and provides the related art. The paper also describes the experimental set-up quite clearly and the results obtained and the discussion on that.

The main drawback of the paper seems to be about its generality. The proposed approach seems to be mainly useful when most of the sampled clients for a given round will have very similar data distribution which is skewed!! I am not sure how probable this scenario is given  a purely random selection of clients in non-iid set-up. I would like to see at least a discussion on this and preferably some analysis on this aspect. The experiments are also done with only a fixed set of 10 clients in each round which is a comparably small number. It will be useful to see how the improvement scale as we increase the number of clients per round for different values of \alpha in Dirichlet distribution.

Other minor comments are below..

- results of FedAvg on CIFAR-10 seems to be lower than state-of-the-art, please check
- rather than reducing the learning rate and using all the gradients, a sampling/weighting of gradients can be done based on their directional heterogeneity.  Some experiments and discussion among these different approaches will be helpful.

**Summary Of The Paper:**

This paper proposed a technique to perform federated learning in the presence of non-IID data across clients. The main claim of the paper is to adjust the learning rate at the server side depending upon the similarity of the skewed gradients from non-iid data clients. The authors claim to achieve higher accuracy as compared to earlier techniques through empirical results.

**Summary Of The Review:**

see above.

---

> ### Author Response · Authors · 2022-11-15
> **Our method is a general approach that aims to address the negative effect of client sampling on federated learning**
>
> **Q1**: Regarding the concern that the situation that most of the sampled clients for a given round will have very similar but skewed data distribution is unlikely to happen and this will affect the generality of our method.
>
> **A1**: **Our method is a general approach that aims to address the negative effect of client sampling when it is applied in federated learning on non-i.i.d. data**. Though the situation in our original motivation that all sampled clients have similar but very skewed distributions may not happen frequently in real cases, it is taken as an example to illustrate that we should set dynamic server learning rates (larger server learning rates for relatively reliable sampling cases, smaller server learning rates when bad sampling happens) based on different client sampling results to stabilize training. **In our main experiments, the sampling strategy is the uniform sampling following realistic situations, and the results show our algorithm can consistently bring improvement in these realistic scenarios**.
>
> Furthermore, we also conducted a special experiment in which we manually decide the bad sampling results in only one specific round, and the results in Figure 4 show that **our method can effectively stabilize the training when the extremely bad client sampling scenario happens even only in one round, which may happen in real cases**.
>
>
> **Q2**: Regarding the experiments with different numbers of clients per round and different values of $\alpha$.
>
> **A2**: **We have conducted experiments with different numbers of clients and with different $\alpha$ in our original submission**. The results and analysis are in Section 5.2 and Appendix L separately, please refer to the corresponding sections for more information. Also, the sampling ratio 0.1 in our main experiments is a practical value under the cross-devices setting (i.e., the number of local devices is huge, such as mobile phones), previous studies have also used the same sampling ratio or even smaller sampling ratios for experiments [1][2][3].
>
> **Q3**: Regarding the results of FedAvg on CIFAR-10.
>
> **A3**: The main reason is that all detailed experimental settings (e.g., data partitioning results, training details, backbones, communication rounds) are not totally the same across our paper and other different papers, but we try to use the most reasonable setting. The reasons for the choices of training hyper-parameters can be found in Appendix J. As for the number of communication rounds, we choose 500 for CIFAR-10 since the global model can already be converged under $\alpha=1.0$ with $E=500$, so $E=500$ is a good choice to reflect the different impacts of different $\alpha$s on federated learning (refer to Appendix L) . However, in our experiments, **we make sure that all methods follow the same experimental settings, so the results can be fairly compared**. We further conduct experiments on FedAvg under $\alpha=0.1$ with sufficient rounds, the converged test accuracy of FedAvg is **62.33**, and after applying FedGLAD, the accuracy is **63.87**.
>
> **Q4**: Regarding the discussion about those gradient re-weighting mechanisms.
>
> **A4**: We believe our method can also be additive with those gradient re-weighing mechanisms [4]. These re-weighting methods aim to calculate a re-weighted average server gradient $g^{t+1} = \sum\limits_{k \in S^{t}} \alpha_{k}^{t}g_{k}^{t}$ where $\sum\limits_{k \in S^{t}} \alpha_{k}=1$. The intuition is setting larger alpha for more reliable local gradient $g_{k}^{t}$ (i.e., the local dataset is more consistent with the global data distribution). In this case,  we can also calculate the corresponding GSI as $$ GSI^{t}=\sqrt{(\sum\nolimits_{k \in S_{t}} || \alpha_{k} \boldsymbol{g}^{t}_{k} ||^{2} )/   (r  || \boldsymbol{g}^{t} ||^{2})}.$$ Then, we can successfully apply FedGLAD.
>
>
> [1] Reddi, Sashank, et al. "Adaptive Federated Optimization." ICLR 2021.
>
> [2] Li, Tian, et al. "Federated optimization in heterogeneous networks." Proceedings of Machine Learning and Systems 2020.
>
> [3] Acar, Durmus Alp Emre, et al. "Federated learning based on dynamic regularization." ICLR 2021
>
> [4] Yeganeh, Yousef, et al. "Inverse distance aggregation for federated learning with non-iid data." Domain Adaptation and Representation Transfer, and Distributed and Collaborative Learning 2020

---

> ### Author Response · Authors · 2022-12-05
> **Sincerely expecting your further feedback**
>
> Dear Reviewer s9jn,
>
> We sincerely thank you for your time and efforts on the reviewing process. We have addressed all your concerns in our previous response, and we are looking forward to your further feedback. Thank you very much!

---

### Official Review · Reviewer_4pTV · 2022-10-24

**Confidence:** 4
**Correctness:** 1
**Technical Novelty And Significance:** 2
**Empirical Novelty And Significance:** 3
**Recommendation:** 3

**Clarity, Quality, Novelty And Reproducibility:**

I found the writing in this work easy to read and liked the style of presentation. Related work in topics of federated learning is well-described, but note my previous contention concerning related work concerning the analysis of stochastic gradient descent.

There is a minor typo in the abstract in "We find that the negative client sampling will cause the merged data distribution of currently sampled
clients heavily inconsistent" -> "We find that the negative client sampling will cause the merged data distribution of currently sampled
clients to be heavily inconsistent"

**Strength And Weaknesses:**

Overall, I like the experimental evaluation done in Section 4, showing the effect of the proposed modification and the extensive commentary on the experimental evaluation in the appendices and submission of code and supplementary material.

My main point of contention though is with the motivation and derivations from the beginning all the way up to Section 3.2. Fundamentally, the phenomenon that the authors observe, that gradient norms computed over small sample sizes differ from the global gradient norm, is unrelated to federated learning. The same analysis and intuition also applies to normal training with stochastic gradient descent - no connection to FL is necessary.

The submission then essentially derives and re-analyzes the conditions for the convergence of stochastic gradient descent, where indeed, a popular assumption is that the variance of the norm of sample gradients is bounded. SGD, as summarized for example in Bottou 2010 "Large-Scale Machine Learning with Stochastic Gradient Descent". SGD is generally thought to be close to an optimal algorithm, even when gradient norm variance is non-zero, but subsequent adaptations and variations do exist, for example in the extensive literature on variance-reduction methods for SGD.

The submission derives the notion of a "gradient similarity-aware indicator" defined as $\mathbb{E}(||g||^2) / ||\mathbb{E}(g)||^2$, based on my re-annotation of Eq.(9) with the gradient $g$ a random variable. Updates are then reweighted based on dividing the current estimate of this quantity with a historical estimate. It is unclear whether this is an actual reduction in effective gradient norm variance, a part where I found have found a proof essential.

Ultimately, I don't believe this motivation makes sense, and it ignores a large body of work on the properties of stochastic gradient descent. The submission adds further wrinkles and complications with a parameter-wise variant and additional hyperparameters in the running average and the running bounds, all of which I'd like to see more carefully ablated, and the overall method potentially works. I can be convinced that the proposed method is empirically useful, based on the empirical evidence provided in Section 4, but I don't find the submitted evidence compelling that this method works for the reasons stated.


**Summary Of The Paper:**

The submission "When to Trust Aggregated Gradients: Addressing Negative Client Sampling in Federated Learning" describes a re-weighting strategy that can be applied during the optimization of ML models in federated learning. Given the full gradient norm over the entire dataset at some point in training, this strategy reweights the gradient norm of every mini-batch sample to be equal. This is combined with a moving average estimation of the full gradient norm, variants with separate rescaling for parameter-groups and dynamic bounds. This approach is evaluated on a range of datasets and models.

**Summary Of The Review:**

This submission motivates a reweighting strategy for federated learning, but the analysis provided would have to equally apply to conventional stochastic gradient descent where I do not find it convincing. The submission adds further complications and produces evidence that the final algorithm is empirically useful, but I do not find the provided evidence compelling that this is for the reasons stated.

 If I fundamentally misunderstood the submission's intent, I would be interested in re-evaluating my score (and I welcome the authors' response in this regard), but otherwise I do not think this is a well-supported submission.

---

> ### Author Response · Authors · 2022-11-15
> **We think your summary of the paper misunderstands the intent of our paper**
>
> Thank you for your great efforts on reviewing our paper. However, we think your summary of our paper misunderstands the intent and content of our work. Our paper is **not about studying the variance of gradient norms in SGD optimization, but to effectively utilize the information of the directions of local gradients to optimize the server learning rates for improving global model’s performance under federated learning**. To help you produce more accurate reviews, we re-state the studied problem and motivation of our paper in the following.
>
>
> **Our studied problem is to analyze and then address the negative impact of client sampling on federated learning with non-i.i.d. data**.  Client sampling practice is widely adopted to reduce the communication cost between the server and clients by sampling only a small part of clients in each round to update the global model.  As we can expect, the merged data distribution of currently sampled clients will be inconsistent with the merged data distribution of all clients, and this will cause that the averaged gradient under partial client participation will deviate from the ideal gradient if all clients participate in each round, which will further cause the aggregation difficulty.
>
>
> Previous studies aim to improve the local training to make the local gradients more consistent or propose better sampling strategies. However, **our work considers from an orthogonal perspective that we aim to improve the server aggregation phase by optimizing the learning rates for the aggregated server gradients**. Our motivation is to find an optimal server learning rate that scales the current server gradient $g_{t}$ to achieve the nearest distance between it and the true optimal gradient $g^{t}_{c}$ (refer to Eq. (5)). We then theoretically find that the optimal server learning rate is positively related to an indicator (GSI) that measures the normalized dissimilarity between the directions of local gradients, and then use this indicator to dynamically adjust the server learning rates. Thus, our main idea is to set larger server learning rates if the sampling results are more reliable, and decrease the server learning rates when bad sampling results happen.
>
> **We sincerely hope you can carefully read our response and re-evaluate our paper. If you have other questions, we would like to have further discussions with you as soon as possible**. Thank you very much!

---

> > ### Comment · Reviewer_4pTV · 2022-11-15
> > **My Concern is not with the Intent, but The Method.**
> >
> > Thank you for your response. To reiterate, my concern is not with the intent of the paper. I am acutely aware that the intent of the paper is to study client sampling in federated learning.
> >
> > My concern is that the proposed method is an accidental redevelopment of classical results in SGD optimization. The reason for this is that the submission uses no assumptions that would be specific to non-IID federated learning. I think this submission would be strengthened by taking a step back and looking at the bigger picture in this way and I would appreciate if the authors could re-read my review with this concern in mind.

---

> > > ### Author Response · Authors · 2022-11-16
> > > **Thank you for your further feedback, but our method is specifically proposed for non-i.i.d. federated learning**
> > >
> > > (1) **Our method is specifically proposed for federated learning on non-i.i.d. data, since the negative effect of client sampling no longer exists if the local data follows i.i.d. partitioning and our proposed GSI is then useless in this case.** That is because there is a large difference between non-i.i.d. federated learning and large-scale centralized SGD optimization.
> > >
> > > In centralized SGD optimization, the samples in each mini-batch are uniformly sampled from the whole dataset, thus **the data distributions across different mini-batches are similar that can be considered as the i.i.d. partitioning case**. Therefore, the gradients in different mini-batches are highly consistent, so our calculated GSI is stable (almost equal to 1) during the training and will not work to adjust the scale of the aggregated gradient.
> > >
> > > However, in federated learning, **the local data distribution is already decided by its local environment, and follows the non-i.i.d. partitioning**. In this case, **the *client sampling* represents sampling each local data distribution rather than sampling each mini-batch in local datasets**. Thus, the sampled local gradients will be divergent caused by different local data distributions, and the aggregated server gradient will also deviate form the true optimal gradient that is assumed to be averaged by all local gradients from whole group of clients. The reliability of aggregated server gradient depends on the meted data distribution of currently sampled clients. Our method then aims to re-weight the scale of the aggregated server gradient based on the reliability of current sampling results. However, we don't aim to achieve that the gradient norms of aggregated server gradients across different  rounds should be the same. Thus, **our method indeed specifically works in the federated learning setting with non-i.i.d. data partitioning.**
> > >
> > > (2) Also, we want to address some misunderstandings in your previous review in the following:
> > >
> > > **M1**:
> > > > "this strategy reweights the gradient norm of every mini-batch sample to be equal"
> > >
> > > **A1**: **Our method will not perform extra operations on each local gradient**, and we follow previous studies' aggregation mechanisms to get the aggregated server gradients first. Our method will then adjust the server learning rate for the aggregated server gradient, i.e., scale the norm of the final aggregated server gradient. Also, **we don't aim to achieve that the gradient norms of aggregated server gradients across different  rounds should be the same.**
> > >
> > >
> > > **M2**:
> > > > "gradient norms computed over small sample sizes differ from the global gradient norm"
> > >
> > > **A2**: Our claim is that the directions of different local gradients are divergent and the reliability of the direction of the aggregated server gradient depends on the sampling results, rather than that the gradient norms of different local gradients differ from the norm of aggregated gradient.
> > >
> > > **M3**:
> > > > "It is unclear whether this is an actual reduction in effective gradient norm variance"
> > >
> > > **A3**: As we mentioned, our paper is not about studying the variance of gradient norms in SGD optimization, as there is a large difference between large-scale centralized SGD optimization and federated learning on non-i.i.d. data.

---

> > > > ### Comment · Reviewer_4pTV · 2022-11-18
> > > > **Response**
> > > >
> > > > Thank you for taking the time to clarify these points. I do appreciate the authors' response.
> > > >
> > > > To reiterate my point though: I do not disagree with your explanation of the importance of non-iid data distributions across clients. My point is that this is not reflected in the motivation and derivations. One could replace every mention of gradient in Sec.2 with "mini-batch gradient sample" (as in classical stochastic gradient descent) and arrive at the same conclusion.

---

> > > > > ### Author Response · Authors · 2022-11-19
> > > > > **Our motivation and derivations are specific to the situation of non-i.i.d. data distributions**
> > > > >
> > > > > **Motivation**:
> > > > >
> > > > > As we mentioned in Section 1 and Figure 1, when the data is non-i.i.d., **the direction of $g_{k}^{t}$ is divergent with each other**, so the aggregated server gradient under partition client participation $g^{t}=\sum\limits_{k \in S^{t}} g_{k}^{t}$ deviates from the ideal gradient $g_{c}^{t} = \sum\limits_{k=1}^{N} g_{k}^{t}$ under full client participation, and the direction of $g^{t}$ relies on the current sampling results. Then, **we are motivated to minimize the distance between $g^{t}$ and $g_{c}^{t}$ in Section 3.2 and Figure 2, in order to avoid updating with $g^{t}$ too much towards a skewed direction**.
> > > > >
> > > > > However, if the data is i.i.d. or in the centralized SGD optimization (in which the mini-batch is i.i.d.), **$g_{k}^{t}$ is consistent with each other, so $g^{t} = g_{c}^{t} $, there is no need to adjust the scale of $g^{t}$ in this case**.
> > > > >
> > > > >
> > > > > **Derivations**:
> > > > >
> > > > > Then we made necessary derivations in Eq. (8) to find an indicator GSI for adjusting the server learning rates. **Though the gradient $g_{k}^{t}$ in Eq. (8) can be any local gradient or mini-batch gradient, the GSI is specifically and only useful when data is non-i.i.d.** as we discussed in the last paragraph of Section 3.2. That is, **if data is i.i.d. or in the centralized SGD optimization (in which the mini-batch is i.i.d.), $g_{i}^{t} = g_{j}^{t}$, then $GSI$ is stable and equals to 1 during the training, and the server learning rates will still be the constant in theory.**
> > > > >
> > > > > When data is non-i.i.d., according to the patterns we visualized in Figure 3 and Appendix R, **the GSI indeed varies across the rounds and can effectively reflect the reliability of sampling results**. Thus, GSI can be used to adjust the server learning rates specifically in non-i.i.d. data settings.
> > > > >
> > > > > We hope our response addresses your concerns, and we look forward to your further feedback!

---

> > > > > > ### Comment · Reviewer_4pTV · 2022-12-10
> > > > > > **Details**
> > > > > >
> > > > > >
> > > > > > A few details:
> > > > > >
> > > > > > >  the direction of $g^t_k$ is divergent with each other, so the aggregated server gradient under partition client participation $g^t$  deviates from the ideal gradient $g^t_c$ under full client participation, and the direction of  relies on the current sampling results
> > > > > >
> > > > > > You could equally define this as: The minibatch $g^t$ is a random subsample of elements $g^t_k$ from the distribution of true gradients which has mean $g^t_c$, but is skewed. I know you intend to use this for a novel application in FL, but it makes sense to connect this to existing research.
> > > > > >
> > > > > > The non-iid component is not used in the derivations in Sec.3. On the opposite, the derivations implicitly assume $\langle g^t, g^t_c \rangle \geq 0$ in Eq.(8). This is implied only implicitly, the bound still works, but is very loose when $\eta^t_o < 0$. As such the derivations already only make sense for directions that are positively aligned and not for directions that are divergent with each other.
> > > > > >
> > > > > > Further, choosing $g^t = g^t_c$ is the wrong analogy to SGD, this would imply full-batch gradient descent. Minibatches $g^t$ deviate from the full batch gradient $g^t_c$ in the same way in SGD as here.

---

> > > > > > > ### Author Response · Authors · 2022-12-11
> > > > > > > **Response**
> > > > > > >
> > > > > > > (1)
> > > > > > > > The minibatch $g^{t}$ is a random subsample of elements $g_{k}^{t}$ from the distribution of true gradients which has mean $g_{c}^{t}$, but is skewed.
> > > > > > >
> > > > > > > We agree with the former part of this claim that in centralized training, $g_{k}^{t}$ can be considered as a random sample from the distribution of true gradients that has mean $g_{c}^{t}$, and this is admitted in our previous response. However, as for the latter part, **we reiterate that since each mini-batch in centralized training is i.i.d., so $g^{t}$ is not skewed from $g_{c}^{t}$ (cosine similarity is still close to 1).**
> > > > > > >
> > > > > > > (2)
> > > > > > > > The non-iid component is not used in the derivations in Sec.3
> > > > > > >
> > > > > > > When deriving Eq. (8), we do not have the non-i.i.d. assumption. **However, when analyzing and interpreting the properties of Eq. (8), we do make the claims under the non-i.i.d. assumption.** That is, $GSI$ indicator is meaningful indicator only when data is non-i.i.d.. Thus, our method is specific to non-i.i.d. federated learning.
> > > > > > >
> > > > > > > (3)
> > > > > > > > As such the derivations already only make sense for directions that are positively aligned and not for directions that are divergent with each other.
> > > > > > >
> > > > > > > Our assumption is that, **the positively aligned variables are $g^{t}$ and $g_{c}^{c}$, and the divergent variables are each $g_{k}^{t}$. They are not contradictory.** ($g_{k}^{t}$ has divergent direction, then the cosine similarity between aggregated gradient $g^{t}$ and $g_{c}^{t}$ is much smaller1 but greater than 0, as the update direction is positive.)
> > > > > > >
> > > > > > > (4)
> > > > > > > >  Further, choosing $g^{t}=g_{c}^{t}$ is the wrong analogy to SGD
> > > > > > >
> > > > > > > We do not intend to relate to the centralized training. In our setting, **$g_{c}^{t}$ is not the full-batch gradient, but the averaged gradient when all clients participate in each round.** Our motivation is (also validated in all previous studies), federated learning behaves better under full client participation than that under partial client participation.

---

> ### Author Response · Authors · 2022-12-05
> **Sincerely expecting your further feedback**
>
> Dear Reviewer 4pTV,
>
> We sincerely thank you for your efforts on reviewing our paper and your corresponding replies on our early responses. We have made detailed clarifications about our motivation and derivations in our last response, and pointed out that **our studied problem and the proposed method are specific for federated learning on non-i.i.d. data rather than for well-shuffled large-batch centralized training.**  That is, **if in the centralized SGD optimization (in which the data across mini-batches is i.i.d.), $g_{i}^{t} = g_{j}^{t}$, then $GSI$ is stable and equals to 1 during the training, and the server learning rates will be the constant in theory (there is no need to adjust the server learning rates in this case).** While in non-i.i.d. federated learning, **$g_{i}^{t}$ varies greatly across local clients, then $GSI$ is a dynamic and meaningful indicator to effectively reflect the reliability of the client sampling results**, and we have provided the evidence in Section 3.2, Section 4.4 and Appendix R.
>
> We sincerely hope you can re-evaluate our method according to our latest responses, and we are looking forward to your further feedback on our responses. Thank you very much!

---

### Official Review · Reviewer_9Lj6 · 2022-10-27

**Confidence:** 4
**Correctness:** 3
**Technical Novelty And Significance:** 2
**Empirical Novelty And Significance:** 3
**Recommendation:** 5

**Clarity, Quality, Novelty And Reproducibility:**

**Clarity:** The paper was easy to follow and clear, with a few exceptions in the methods section 3.2 as discussed above.

**Novelty:** I find the idea of using gradient similarity to determine the learning rate to be particularly novel.

**Reproducibility:** The authors have made sufficient efforts to ensure reproducibility.

**Strength And Weaknesses:**

**Strengths**

**S1** The paper addresses a challenging and highly relevant problem in federated learning w/ non-iid client distributions.

**S2** The experiments in the non-iid setting are exhaustive with plenty of ablations. Moreover, the proposed FedGLAD outperforms commonly used baselines such as FedAvg, FedProx and FedAdam in the non-iid federated learning settings.

**S3** The paper is clear and easy to follow, (w/ some exceptions as discussed below).

**Weaknesses**

**W1** The theoretical motivation in section 3.2 and the corresponding claim that FedGLAD can mitigate the inconsistency b/w $g_t$ and $g_c$ doesn't seem solid. The technical motivation is inconsistent with the final method adopted (Please see the review justification).

**W2** I think the idea of using GSI to determine the learning rate is interesting and clever. But with increasing the learning rate for dissimilar client gradients, the method currently assumes that dissimilar gradients can "only" reflect useful global gradient. It seems only using non-iid partitions in the experiments may not reflect other weaknesses of this approach. For e.g. in the noisy federated learning setting [1] gradients will likely be dissimilar due to the noise, and increasing the server learning rate could lead to poor performance.   Another federated learning setting to consider could be the multi-modal non-iid setting in Federated Learning [2]. I think there should at least be a discussion on how FedGLAD would do in these settings.

**References**

[1] Robust Federated Learning with Noisy Labels, Yang et al. (2020)

[2] Weight Anonymized Factorization for Federated Learning, Hao et al. (2020)


**Summary Of The Paper:**

The paper proposes a method to improve the server optimization for non-iid settings in federated learning. The proposed method involves using the client gradients to adapt the server learning rate such that similar gradients should have a low learning rate, while dissimilar gradients will have a high server learning rate. The approach builds upon the intuition that similar gradients may correspond to a biased average gradient from similar client distributions, which may not correspond to the global average gradient if all the clients were used. The authors validate their proposal with several baselines on 4 datasets and demonstrate encouraging results.

**Summary Of The Review:**

My main concerns are listed below --

- In equation 8, since $||g_c||$ is considered a constant, the learning rate bound is not particularly unique for Equation 5. This bound will also hold for any constant vector in place of $g_c$, since it will be dropped later? The authors should clarify how the introduced scaling in $g$ can make it closer particularly to $g_c$ (As has been claimed in the third paragraph of section 3.2).

- The paper also refers to Fig 1 and Fig 2 in section 3.2 to illustrate the motivation of Equation 5. But as discussed above, the GSI-based scaling is not particularly solving equation 5. Again, the claim of mitigating the inconsistency b/w $g_t$ and $g_c$ is not well-supported.

- While the authors motivate the non-iid setting from real-world data, other settings such as noisy labels [1] and client imbalance [2] are not considered. I feel the GSI-based scaling makes strong assumptions that the non-iid setting is the only challenge in the data distributions and the experiments are also designed in this isolated setting. Note that all the other baselines do not strictly make any such assumption. The paper should at least discuss the applicability of FedGLAD in other real-world federated learning settings.

---

> ### Author Response · Authors · 2022-11-15
> **Thank you for your questions**
>
> **Q1**: Regarding the clarification why the introduced scaling in $g^{t}$ can make it closer particularly to $g_{c}^{t}$.
>
> **A1**: The main point is, **the exponential average of historical GSIs $B^{t}$ can be considered as an estimator to estimate the value of GSI under the full client participation, and calculating the ratio between current $GSI^{t}$ and $B^{t}$ for $\eta^{t}$ indeed aims to scale the $g^{t}$ closer particularly to $g^{t}_{c}$**.
>
> The reason is, Eq. (8) can be further extended as
> $$\eta_{o}^{t}  \leq \frac{  \eta_{s}  || g_{c}^{t} ||}{\sqrt{\frac{1}{r} \sum_{k \in S_{t}}|| g_{k}^{t} ||^{2}  }} \frac{\sqrt{\frac{1}{r} \sum_{k \in S_{t}}|| g_{k}^{t} ||^{2}  }}{|| g^{t} ||}
> =\eta_{s}  \frac{\sqrt{\frac{1}{N} \sum_{k =1}^{N}|| g_{k}^{t} ||^{2}  }}{\sqrt{\frac{1}{r} \sum_{k \in S_{t}}|| g_{k}^{t} ||^{2}  }}
> \frac{ || g_{c}^{t} ||}{\sqrt{\frac{1}{N} \sum_{k =1}^{N}|| g_{k}^{t} ||^{2}  }}  \frac{\sqrt{\frac{1}{r} \sum_{k \in S_{t}}|| g_{k}^{t} ||^{2}  }}{|| g^{t} ||}.
> $$
> According to our assumption about the scale component, $  \frac{\sqrt{\frac{1}{N} \sum_{k =1}^{N}|| g_{k}^{t} ||^{2}  }}{\sqrt{\frac{1}{r} \sum_{k \in S_{t}}|| g_{k}^{t} ||^{2}  }}$ is about 1. Then, $\frac{\sqrt{\frac{1}{N} \sum_{k =1}^{N}|| g_{k}^{t} ||^{2}  }}{ || g_{c}^{t} ||}$ is the GSI under the full client participation, and we estimate it with $B^{t}$ in Eq. (11). Therefore, our solution in Eq. (11) indeed scales the $g^{t}$ correctly to satisfy our original goal.
>
> **Q2**: Regarding the applicability of FedGLAD in other real-world federated learning settings.
>
> **A2**: Thank you for pointing out this problem, **we had a discussion about a similar scenario in our Ethics Statement section (at the beginning of Page 10) and we think our method is also applicable in those settings**.
>
> In the Ethics Statement section, we proposed the corresponding countermeasures:
>
> (a) When some local clients are adversarial attackers [1] or the local data is noisy [2], our mechanism may face some problems since our method tends to give larger server learning rates when local gradients have more divergent directions. However, (a.1) as for the adversarial clients, **we can apply existing detecting techniques [1] to filter out extremely abnormal gradients first before aggregation**; (a.2) As for the noisy labels problem, previous studies [2] managed to select the most confident samples for local training only, in order to avoid the negative impact of incorrectly labeled samples. Therefore, our method is also orthogonal to them since we only make the improvement in the server side, so **our method can be combined with them to deal with the federated learning with noisy labels**.
>
> (b) As for another reference you mentioned [3], the “multi-modal” non-i.i.d. setting means the merged data distribution of all clients (global data distribution) is unbalanced. As we mentioned in our main paper (refer to Footnote 1), **our method does not have the assumption on the global data distribution, so our method is also applicable in this scenario**. We further conduct the experiments to verify the effectiveness of FedGLAD when global data distribution is imbalanced. We keep the global data distribution the same as that in [3] (i.e., 5 of 10 categories have five times more sample size than other 5 categories), and other settings are the same as that in our main experiments. The results are in the following Table 1. As we can see, **FedGLAD can also bring improvement when the global data distribution is imbalanced**.
>
> Table 1. The results in the settings where the global data distributions are imbalanced.
>
> | Dataset | Method | Test Acc. (%) |
> | :-----| :---- | :----: |
> | CIFAR-10 | FedAvg | 30.28 |
> | CIFAR-10 | FedAvg+FedGLAD | **31.55** |
> | MNIST | FedAvg | 65.75 |
> | MNIST | FedAvg+FedGLAD | **67.06** |
>
> [1] Blanchard, Peva, et al. "Machine learning with adversaries: Byzantine tolerant gradient descent." NIPS 2017
>
> [2] Yang, Seunghan, et al. "Robust federated learning with noisy labels." IEEE Intelligent Systems 2022
>
> [3] Hao, Weituo, et al. "Waffle: Weight anonymized factorization for federated learning." IEEE Access 2022

---

> > ### Comment · Reviewer_9Lj6 · 2022-12-06
> > **Acknowledgment of the rebuttal**
> >
> > I thank the authors for their response. I appreciate the additional results provided in the response. However, I am afraid that my concerns regarding section 3.2 still hold. The explanation provided by the authors is not convincing in why scaled $g_t$ is closer particularly to $g^t_c$. If $g^t_c$ is considered a constant and dropped in the analysis, I don't think the claim holds.
> >
> > I have also read the response from other reviewers, and I agree with reviewer 4pTV regarding the disconnect in the motivation (section 3.2) and the final derived results.

---

> > > ### Author Response · Authors · 2022-12-06
> > > **Thank you for your feedback**
> > >
> > > Thank you for your feedback on our response. Regarding your remaining concern on "*why scaled $g^{t}$ is closer particularly to $g_{c}^{t}$*", we make the detailed clarifications in the following:
> > >
> > > The main point is, **we didn't drop $|| g_{c}^{t}||$ when deriving our method. Instead, we try to estimate it**.
> > >
> > > (1) First of all, in Eq. (8), we get $\eta_{t} \leq \eta_{s} \frac{|| g_{c}^{t}||}{|| g^{t}||}$, and as we derived in our previous response, we can extend the Eq. (8) as
> > >
> > > $$\eta_{o}^{t}  \leq
> > > \eta_{s}  \frac{\sqrt{\frac{1}{N} \sum_{k =1}^{N}|| g_{k}^{t} ||^{2}  }}{\sqrt{\frac{1}{r} \sum_{k \in S_{t}}|| g_{k}^{t} ||^{2}  }}
> > > \frac{ || g_{c}^{t} ||}{\sqrt{\frac{1}{N} \sum_{k =1}^{N}|| g_{k}^{t} ||^{2}  }}  \frac{\sqrt{\frac{1}{r} \sum_{k \in S_{t}}|| g_{k}^{t} ||^{2}  }}{|| g^{t} ||} = \eta_{s}  \frac{ || g_{c}^{t} ||}{\sqrt{\frac{1}{N} \sum_{k =1}^{N}|| g_{k}^{t} ||^{2}  }} GSI^{t}.
> > > $$
> > > In the above inequality, $\frac{ || g_{c}^{t} ||}{\sqrt{\frac{1}{N} \sum_{k =1}^{N}|| g_{k}^{t} ||^{2}  }}$ is the $GSI$ under the setting when all clients participate in each round, which is also an unknown constant. Now, **if we assume the value of $\frac{ || g_{c}^{t} ||}{\sqrt{\frac{1}{N} \sum_{k =1}^{N}|| g_{k}^{t} ||^{2}  }}$ is known, then we can use the $GSI$ indicator and the value of $\frac{ || g_{c}^{t} ||}{\sqrt{\frac{1}{N} \sum_{k =1}^{N}|| g_{k}^{t} ||^{2}  }}$ to re-scale $g^{t}$ to make it closer to $g^{t}_{c}$.**
> > >
> > > (2) However, the remaining problem is that, $\frac{ || g_{c}^{t} ||}{\sqrt{\frac{1}{N} \sum_{k =1}^{N}|| g_{k}^{t} ||^{2}  }}$ is unknown.  **Then, our solution on dealing with unknown $\frac{ || g_{c}^{t} ||}{\sqrt{\frac{1}{N} \sum_{k =1}^{N}|| g_{k}^{t} ||^{2}  }}$ is to estimate it by using an estimator $B^{t}$ that contains the information from historical $GSI$s in Eq. (10).**
> > >
> > > (3) Finally, combining the analysis in (1) and (2), our proposed method in Eq. (11) particularly aims to scale $g^{t}$ closer to $g_{c}^{t}$.

---

> ### Author Response · Authors · 2022-12-05
> **Sincerely expecting your further feedback**
>
> Dear Reviewer 9Lj6,
>
> We thank you for your efforts on the reviewing process. We have responded to all your questions in our previous response. As the author-reviewer discussion deadline is approaching, we sincerely expect your further feedback on our response. Thank you very much!

---

### Author Response · Authors · 2022-11-15
**General Response**

We sincerely thank all the reviewers for their precious time and great efforts on the reviewing process, and their constructive suggestions! We are glad that they think our studied problem is important (Reviewer 9Lj6) and we provide some novel insights  (Reviewer 9Lj6, Reviewer s9jn and Reviewer VDBr). We are pleased that they agree that our paper is well-written (all four reviewers) and our experiments are well-designed and thorough (all four reviewers). To help produce more accurate reviews, we address the questions and doubts raised by each reviewer below. We sincerely look forward to having further discussions if they have other questions.

---

### Author Response · Authors · 2022-12-02
**Sincerely expecting the further feedback**

Dear all reviewers and Area Chairs,

We have addressed all the questions and concerns of each reviewer in the previous responses. As the author-reviewer discussion deadline is approaching, we sincerely expect the further feedback from all reviewers. Thank you very much!

---

> ### Author Response · Authors · 2022-12-05
> **Summary of the updated revision**
>
> Dear all reviewers,
>
> In our latest revision, we follow the constructive suggestion from Reviewer VDBr and add a new section in the Appendix (**refer to Appendix R**) to support our claim that GSI can reflect the distribution discrepancy between the merged dataset of selected clients and the merged dataset of all clients.
>
> We make the visualizations about the relationship between the GSI and the similarity between the merged data distribution of selected clients and that of all clients. The main conclusion is, **GSI is positively related to the merged data distribution similarity between the sampled clients and all clients, and can effectively reflect the sampling results.**

---

### Decision · Program_Chairs · 2023-01-20

**Decision:**

Reject

**Justification For Why Not Higher Score:**

Theoretical motivation is unclear and not rigorous enough. Further no convergence rate analysis is provided to show the adaptive mechanism improves the rate.

**Justification For Why Not Lower Score:**

N/A

**Metareview: Summary, Strengths And Weaknesses:**

Paper studies Federated Learning and notes that the sampled average client pseudo-gradients $g^t$ may not align with global gradient $g^t_c$ if the sample of clients are biased. This motivates them to study an adaptive server learning rate mechanism which aims to make the gradient step $\eta_t g^t$ as close to ideal global server step $\eta_s g^t_c$ as possible. Paper provides some good results and ablation studies over many base FL algorithms and datasets. The parameter group-wise adaptation was also very interesting. Reviewers noted that the theoretical motivation for mechanism was not rigorous enough and may be making strong assumptions. Further no convergence rate guarantees are provided even for simple settings. Paper clarity would also improve if the authors can flesh out the connections to the vast literature on adaptive learning rates for centralized optimization. A good future experiment to solidify the theoretical motivation of the paper would be to verify that the performance increases when using the ideal learning rate $\eta_o^t$ claimed in equation (7), at least for smaller-scale problems (of course this needs computing. $g^t_c$). Authors can also do a correlation study between this ideal lr and the estimated lr in the algorithm. Finally, it might be better to term GSI as dissimilarity to avoid confusion since it at minimum when client pseudo gradients are the same.